# A regulatory module controlling stress-induced cell cycle arrest in *Arabidopsis*

**Naoki Takahashi[1], Nobuo Ogita[1], Tomonobu Takahashi[1], Shoji Taniguchi[1], Maho Tanaka[2,3], Motoaki Seki[2,3], Masaaki Umeda[1]\***

[1]Graduate School of Science and Technology, Nara Institute of Science and Technology, Nara, Japan; [2]RIKEN Center for Sustainable Resource Science, Yokohama, Japan; [3]RIKEN Cluster for Pioneering Research, Wako, Japan

**Abstract** Cell cycle arrest is an active response to stresses that enables organisms to survive under fluctuating environmental conditions. While signalling pathways that inhibit cell cycle progression have been elucidated, the putative core module orchestrating cell cycle arrest in response to various stresses is still elusive. Here we report that in *Arabidopsis*, the NAC-type transcription factors ANAC044 and ANAC085 are required for DNA damage-induced G2 arrest. Under genotoxic stress conditions, ANAC044 and ANAC085 enhance protein accumulation of the R1R2R3-type Myb transcription factor (Rep-MYB), which represses G2/M-specific genes. ANAC044/ANAC085-dependent accumulation of Rep-MYB and cell cycle arrest are also observed in the response to heat stress that causes G2 arrest, but not to osmotic stress that retards G1 progression. These results suggest that plants deploy the ANAC044/ANAC085-mediated signalling module as a hub which perceives distinct stress signals and leads to G2 arrest.

DOI: https://doi.org/10.7554/eLife.43944.001

## Introduction

To survive under fluctuating environmental conditions, all organisms have the ability to deal with various kinds of internal and external stresses. In animals, failure to adapt against disadvantageous stresses leads to various diseases such as cancer, and, in severe cases, cells perish (*Twayana and Ravanan, 2018*). To avoid such consequences, cells deploy mechanisms to temporarily arrest proliferation, thereby ensuring appropriate stress responses. For instance, the high osmolarity glycerol (HOG) MAP kinase (MAPK) cascade induces cell cycle arrest by inhibiting the transcription of cyclin genes in response to hyperosmotic stress (*Saito and Posas, 2012*). However, while such individual pathways inducing inhibition of cell proliferation have been well characterized, our understanding of the central module that is presumed to govern cell cycle arrest in response to different stresses is still limited. Due to their sessile lifestyle, plants also need to handle many kinds of abiotic and biotic stresses, and to adapt themselves to environmental conditions by activating particular signalling cascades. Various transcription factors that respond to abiotic stresses have been well studied, such as those belonging to the MYB, bZIP, drought-responsive element-binding protein (DREB) and DREB1s/C-repeat binding factor (CBF) families (*Vinocur and Altman, 2005*; *Umezawa et al., 2006*; *Golldack et al., 2011*). However, downstream signalling pathways that inhibit cell proliferation remain largely unknown; therefore, it remains to be established whether plants deploy central signalling pathway(s) that induce cell cycle arrest upon exposure to multiple stresses.

DNA damage also arrests the cell cycle to allow DNA repair to occur before DNA replication or mitosis begins (*Harper and Elledge, 2007*; *Ciccia and Elledge, 2010*; *Hu et al., 2016*). The programmed response to DNA damage is therefore crucial to maintain genome integrity. DNA damage is sensed by two kinases, ATAXIA TELANGIECTASIA MUTATED (ATM) and ATM AND RAD3-RELATED (ATR) (*Abraham, 2001*; *Garcia et al., 2003*; *Culligan et al., 2004*). ATM is activated by

\*For correspondence:
mumeda@bs.naist.jp

**Competing interests:** The authors declare that no competing interests exist.

**eLife digest** During environmental stresses, such as high light or a drought, plants do not have the opportunity to up and leave. Instead, they need to buy time and energy by pausing their growth, which means stopping their cells from dividing. In this case, the cell cycle, a series of stages during which a cell prepares itself for division, must be halted.

If the genetic information in cells is damaged, often under the influence of the environment, plants stop their cell cycle in the step just before division. However, it is still unclear how this process takes place, and which proteins participate in it. Researchers also do not know whether environmental stresses can directly trigger this mechanism.

To investigate, Takahashi et al. conducted a series of genetic experiments on a common plant known as *Arabidopsis thaliana*, and they identified two proteins, ANAC044 and ANAC085, which could stop the cell cycle when the genetic information is damaged. In particular, ANAC044 and ANAC085 worked by stabilizing other proteins that turn off certain genes that the cell needed to divide. Additional experiments showed that other types of stresses, such as heat, halted the cell cycle using the ANAC044 and ANAC085 pathway. This suggests that this mechanism may be a central 'hub' that responds to various stress signals from the environment to prevent cells from dividing.

In the field, environmental stresses stop plants from growing, which reduces crop yields; ultimately, manipulating ANAC044 or ANAC085 might help to boost plant productivity even when external conditions fluctuate.

DOI: https://doi.org/10.7554/eLife.43944.002

double-strand breaks (DSBs), whereas ATR senses single-strand breaks (SSBs) and stalled replication forks (*Bensimon et al., 2011*; *Flynn and Zou, 2011*). In animals, ATM phosphorylates the checkpoint kinase CHK2, which stabilizes the transcription factor p53 and induces expression of the CDK inhibitor p21, resulting in cell cycle arrest at G1/S phase (*Donjerkovic and Scott, 2000*; *Bartek and Lukas, 2001*; *Cheng and Chen, 2010*). CHK1, which is phosphorylated by ATR, and CHK2 also activate the G2/M-phase checkpoint by phosphorylating CDC25 phosphatases (*Karlsson-Rosenthal and Millar, 2006*; *Perry and Kornbluth, 2007*). In contrast, while plants possess ATM and ATR, they lack functional orthologues for p21, p53, CDC25 or CHK1/2. Instead, a plant-specific NAC-type transcription factor, SUPPRESSOR OF GAMMA RESPONSE 1 (SOG1), is phosphorylated and activated by ATM and ATR, and triggers DNA damage responses (DDRs) (*Yoshiyama et al., 2009*; *Yoshiyama et al., 2013*; *Sjogren et al., 2015*; *Yoshiyama et al., 2017*).

A previous study demonstrated that in *Arabidopsis*, SOG1 regulates the expression of almost all genes induced by DSBs (*Yoshiyama et al., 2009*), causes G2 arrest in the meristem, enhances the transition from cell division to endoreplication (a repeated cycle of DNA replication without mitosis), and triggers stem cell death (*Furukawa et al., 2010*; *Adachi et al., 2011*). Expression of CDK inhibitors, such as *SMR5* and *SMR7*, is directly induced by SOG1 (*Yin et al., 2014*; *Ogita et al., 2018*); however, their induction also occurs in DSB-tolerant *Arabidopsis* mutants, suggesting that it is not sufficient to arrest the cell cycle (*Chen et al., 2017*). We previously demonstrated that simultaneous suppression of a set of G2/M-specific genes is accompanied by DNA damage-induced G2 arrest (*Adachi et al., 2011*). G2/M-specific genes, such as those encoding mitotic cyclins, are controlled by three Myb repeat-containing transcription factors, called R1R2R3-Myb or MYB3R (*Ito, 2005*). *Arabidopsis* possesses five genes for MYB3R, MYB3R1 to MYB3R5, among which MYB3R4 acts as a transcriptional activator (Act-MYB), and MYB3R3 and MYB3R5 are transcriptional repressors (Rep-MYB) (*Haga et al., 2007*; *Haga et al., 2011*; *Kobayashi et al., 2015*). MYB3R1 functions as both an activator and a repressor (*Kobayashi et al., 2015*). MYB3Rs bind to target gene promoters via a *cis*-acting element called the mitosis-specific activator (MSA) element, and control the expression of G2/M-specific genes (*Kobayashi et al., 2015*). We recently reported that Rep-MYBs are actively degraded via the ubiquitin-proteasome pathway, but become stabilized under DNA damage conditions, thereby suppressing G2/M-specific genes and causing G2 arrest (*Chen et al., 2017*). Protein stability of Rep-MYBs is dependent on CDK activity: CDK phosphorylates the C-terminal domain of MYB3R3/5 and promotes their degradation. Therefore, we proposed a model in which DNA damage

phosphorylates and activates SOG1, and reduces CDK activities, probably by inducing CDK inhibitor genes, thereby stabilizing Rep-MYBs (**Chen et al., 2017**). Since Rep-MYBs repress mitotic cyclin expression, their stabilization further decreases CDK activities, thus resulting in cell cycle arrest. A recent report also demonstrated that Rep-MYBs function as major repressors of cell cycle genes in the response to DNA damage (**Bourbousse et al., 2018**). However, it remains unknown whether the initial reduction of CDK activity is sufficient to trigger Rep-MYB stabilization, or whether another signalling pathway is required to accumulate a sufficient amount of Rep-MYB for the suppression of numerous G2/M-specific genes.

In this study, we show that the NAC-type transcription factors ANAC044 and ANAC085, which are direct targets of SOG1, play a crucial role in causing G2 arrest in response to DNA damage. In the absence of ANAC044 or ANAC085, Rep-MYB does not accumulate and cannot inhibit G2/M progression, and the plants exhibit DNA damage tolerance. ANAC044 and ANAC085 are not required for the induction of genes for DNA repair proteins or CDK inhibitors, but rather are engaged in accumulation of Rep-MYBs. Our data show that they also participate in heat stress-induced inhibition of G2 progression, suggesting that an ANAC044- and ANAC085-mediated pathway plays a central role in orchestrating stress-induced G2 arrest by controlling Rep-MYB accumulation.

## Results

### ANAC044 and ANAC085 are induced by DNA damage

Previous studies demonstrated that the SOG1 transcription factor controls cell cycle arrest and stem cell death (**Yoshiyama et al., 2009**; **Furukawa et al., 2010**; **Adachi et al., 2011**). Therefore, it is conceivable that DNA damage-induced G2 arrest is controlled by signalling pathways downstream of SOG1. Recently we identified 146 *Arabidopsis* genes that are directly targeted by SOG1, among which were the NAC transcription factors ANAC044 and ANAC085 (**Ogita et al., 2018**). Phylogenetic analysis of NAC transcription factors indicated that ANAC044 and ANAC085 are the closest relatives of SOG1 (**Figure 1A**); indeed, in the NAC domain, which is essential for DNA binding, their amino acid similarity to SOG1 is 72.0% for ANAC044 and 72.6% for ANAC085. A striking difference is that the C-terminal regions carrying the domain needed for transcriptional regulation are shorter in ANAC044 and ANAC085 than in SOG1. Moreover, five serine-glutamine (SQ) motifs, which are targets for phosphorylation by ATM and ATR, are present towards the C terminus of SOG1, but missing in ANAC044 and ANAC085 (**Figure 1B**). We therefore predicted that ANAC044 and ANAC085 exert distinct functions in the DDR.

We first examined the transcriptional response of these NAC transcription factors to the DSB-inducing agent bleomycin. Five-day-old wild-type (WT) seedlings were transferred onto MS medium containing 0.6 µg/ml bleomycin, and quantitative real-time PCR (qRT-PCR) was conducted using total RNA isolated from root tips. The transcript levels of *ANAC044* and *ANAC085* increased soon after bleomycin treatment and reached maxima after 12 hr, indicating a rapid response to DSBs (**Figure 1—figure supplement 1A**). We next generated *promoter:GUS* reporter lines and found that *ProANAC085:GUS* showed no GUS signal irrespective of bleomycin treatment, probably due to a lack of regulatory element(s) essential for transcriptional activation in the promoter region. On the other hand, in the *ProANAC044:GUS* line, *GUS* expression was elevated by bleomycin treatment in root tips and young leaves (**Figure 1—figure supplement 1B and C**). Interestingly, we observed a patchy pattern of signals in the root tip, suggesting that *ANAC044* is induced in cells at particular cell cycle stage(s) that are more sensitive to DSBs. We also tested other DNA-damaging agents, namely hydroxyurea (HU), mitomycin C (MMC) and methyl methanesulfonate (MMS). HU inhibits deoxyribonucleotide production, thereby causing DNA replication stress (**Wang and Liu, 2006**; **Saban and Bujak, 2009**). MMS is an alkylating agent that methylates guanine and adenine bases, resulting in base mispairing and replication blocks (**Beranek, 1990**; **Llorente et al., 2008**). MMC generates interstrand cross-links on DNA (**Rink et al., 1996**). Treatment with all three drugs again induced *GUS* expression in root tips and young leaves (**Figure 1—figure supplement 1B and C**), suggesting that ANAC044 is associated with the DDR in actively dividing cells.

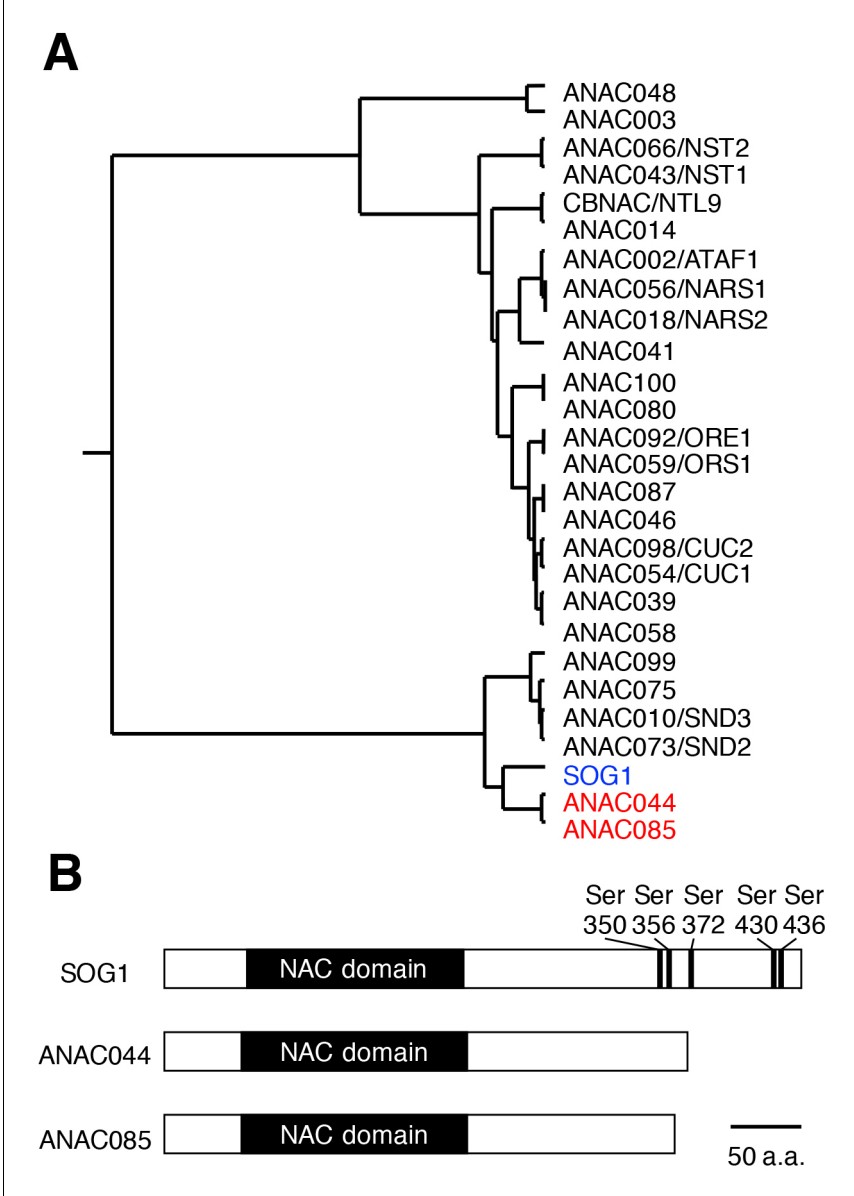

**Figure 1.** Similarities among SOG1, ANAC044 and ANAC085. (**A**) Phylogenetic tree of the NAC transcription factors in *Arabidopsis*. SOG1 (blue letters) and ANAC044 and ANAC085 (red) are highlighted. (**B**) Protein structures of SOG1, ANAC044 and ANAC085. Serine (Ser) residues of the SQ motifs in SOG1 are indicated with numbers of the amino acid sequence.

DOI: https://doi.org/10.7554/eLife.43944.003

The following source data and figure supplements are available for figure 1:

**Figure supplement 1.** *ANAC044* and *ANAC085* are induced by DNA damage.
DOI: https://doi.org/10.7554/eLife.43944.004

**Figure supplement 1—source data 1.** Source data.
DOI: https://doi.org/10.7554/eLife.43944.005

**Figure supplement 2.** SOG1-mediated induction of *ANAC044* and *ANAC085*.
DOI: https://doi.org/10.7554/eLife.43944.006

**Figure supplement 2—source data 1.** Source data.
DOI: https://doi.org/10.7554/eLife.43944.007

## SOG1 directly induces *ANAC044* and *ANAC085* under DNA damage conditions

To examine whether the transcriptional induction of *ANAC044* and *ANAC085* is mediated by SOG1, we conducted qRT-PCR using RNA isolated from the *sog1* knockout mutant *sog1-101* (*Ogita et al., 2018*). As shown in *Figure 1—figure supplement 2A*, bleomycin-induced expression of *ANAC044* and *ANAC085* was not observed in *sog1-101*. A similar result was also obtained using the *ProANAC044:GUS* reporter gene, which did not show bleomycin-triggered *GUS* expression in the *sog1-101* mutant background (*Figure 1—figure supplement 2B*), indicating that SOG1 is required for the transcriptional induction.

As described above, we had identified *ANAC044* and *ANAC085* as SOG1 target genes because their promoters displayed significant peaks in the ChIP-seq data of SOG1 (*Ogita et al., 2018*). We searched for the SOG1-binding motif CTT(N)$_7$AAG in the promoter regions of *ANAC044* and *ANAC085*, and found CTTGGGGAGCAAG and CTTATTTACGAAG sequences, respectively (*Figure 1—figure supplement 2C*). To reveal whether SOG1 binds to the identified promoter regions, we conducted ChIP-qPCR analysis. Using transgenic plants harbouring *ProSOG1:SOG1-Myc*, which can complement the *sog1-1* mutation (*Yoshiyama et al., 2013*), immunoprecipitation was performed with anti-Myc antibody, and chromatin fragments bound to SOG1-Myc were subjected to qPCR. The result showed that for both *ANAC044* and *ANAC085*, the promoter regions containing the SOG1-binding motifs were highly enriched when chromatin isolated from bleomycin-treated plants, but not from non-treated control, was used for immunoprecipitation (*Figure 1—figure supplement 2C*). This strongly suggests that SOG1 binds to the *ANAC044* and *ANAC085* promoters under DNA damage conditions and induces their expression. DNA damage-dependent binding of SOG1 has also been demonstrated for promoters of other SOG1 target genes (*Yin et al., 2014*; *Ogita et al., 2018*).

## *anac044* and *anac085* mutants are tolerant to DNA damage

We then examined the involvement of *ANAC044* and *ANAC085* in the DDR using multiple mutant alleles. *anac044-1* (SAIL_1286D02) and *anac044-2* (GABI_968B05) have T-DNA insertions in the fourth and fifth exons of *ANAC044*, respectively, and *anac085-1* (GABI_894G04) and *anac085-2* (SALK_208662) contain T-DNA fragments in the fourth exon of *ANAC085* (*Figure 2—figure supplement 1A*). Semi-quantitative RT-PCR showed that transcripts of *ANAC044* in *anac044-1* and *anac044-2*, and *ANAC085* in *anac085-1* and *anac085-2*, were undetectable using primers flanking the T-DNA insertion sites, indicating that all are null alleles (*Figure 2—figure supplement 1B*). Five-day-old seedlings grown on MS medium were transferred onto a medium supplemented with or without bleomycin, HU, MMC or MMS, and grown for a further 5 days. Measurement of root growth showed that the two mutants for each of *anac044* or *anac085* were more tolerant than WT to all four DNA-damaging agents, and that root growth was almost the same between *anac044* and *anac085* (*Figure 2—figure supplement 2A*). Moreover, the *anac044-1 anac085-1* double mutant showed a similar tolerance to that of the single mutants (*Figure 2*; *Figure 2—figure supplement 3*).

## ANAC044 and ANAC085 mediate SOG1-dependent DDRs

Previous studies demonstrated that *sog1-1* was tolerant to gamma irradiation and bleomycin treatment, and was hypersensitive to HU (*Yoshiyama et al., 2009*; *Hu et al., 2015*; *Yoshiyama et al., 2017*). We found that the tolerance to bleomycin and MMS was comparable between *sog1-101*, *anac044-1 anac085-1* and the triple mutant *anac044-1 anac085-1 sog1-101* (*Figure 2—figure supplement 2B*). Moreover, the triple mutant exhibited hypersensitivity to HU and MMC to a similar extent to that of *sog1-101* (*Figure 2—figure supplement 2B*), indicating that *sog1-101* is epistatic to *anac044-1 anac085-1*. These results suggest that *ANAC044* and *ANAC085* function downstream of *SOG1* in DNA damage-induced root growth retardation.

It was reported that DSBs induce cell death of stem cells and stele precursor cells in the root tip, and that SOG1 is involved in this response (*Furukawa et al., 2010*). To reveal whether *ANAC044* and *ANAC085* are also required for DSB-induced stem cell death, five-day-old seedlings were treated with bleomycin for 24 hr, and root tips were observed. Propidium iodide (PI)-stained dead cells were visible in the stem cell and stele precursor cell populations in WT, while almost no cell death was observed in *anac044-1 anac085-1*, *sog1-101* or *anac044-1 anac085-1 sog1-101*

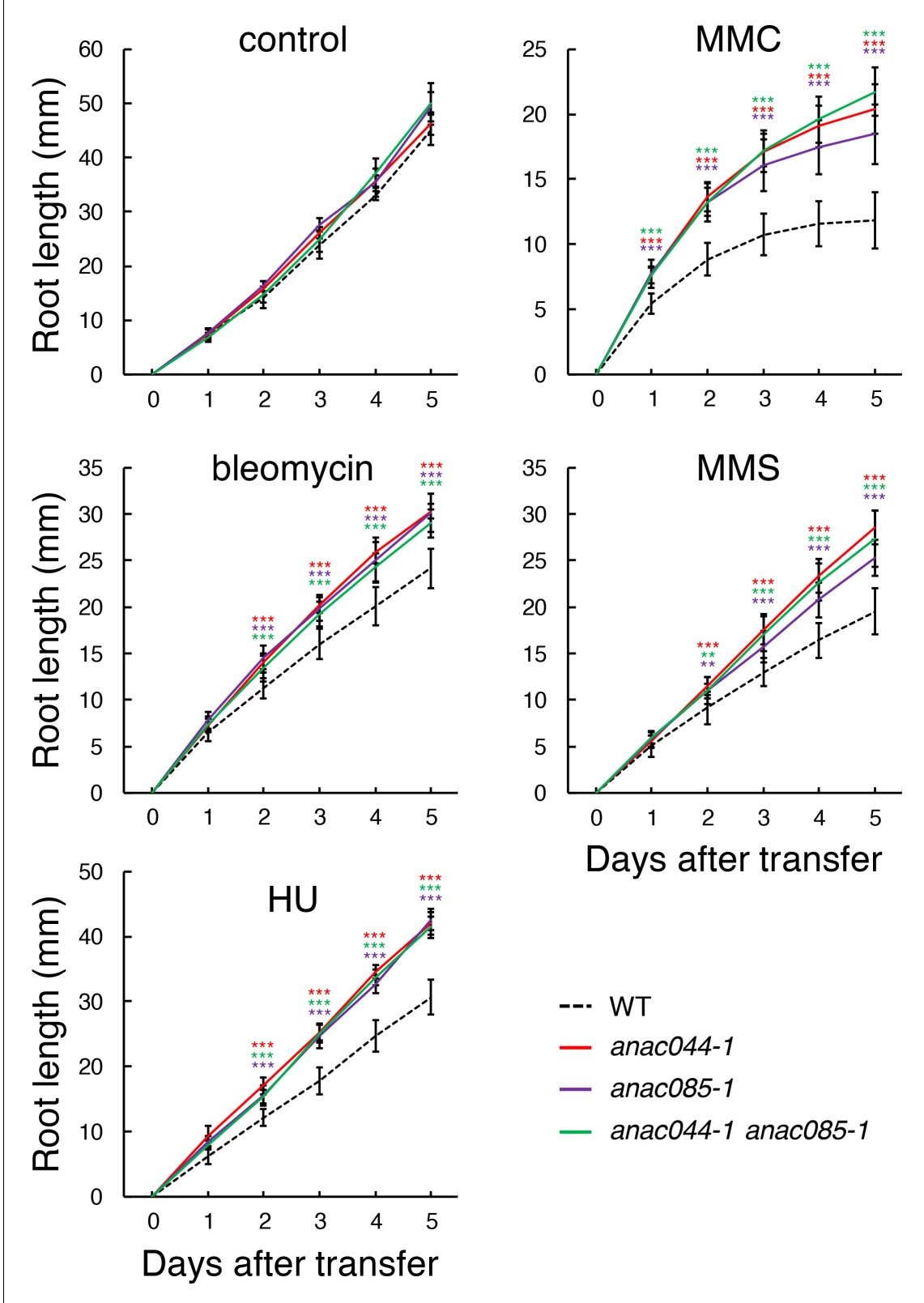

**Figure 2.** *anac044* and *anac085* are tolerant to DNA damage. Five-day-old seedlings of WT, *anac044-1*, *anac085-1* and *anac044-1 anac085-1* were transferred to medium with or without 0.6 µg/ml bleomycin, 1.5 mM hydroxyurea (HU), 3.3 µg/ml mitomycin C (MMC) or 80 ppm methyl methanesulfonate (MMS), and root length was measured every 24 hr. Data are presented as mean ± SD (n = 13). Significant differences from WT were determined by Student's *t*-test: **, p<0.01; ***, p<0.001.

*Figure 2 continued on next page*

*Figure 2 continued*

DOI: https://doi.org/10.7554/eLife.43944.008

The following source data and figure supplements are available for figure 2:

**Source data 1.** Source data.

DOI: https://doi.org/10.7554/eLife.43944.013

**Figure supplement 1.** T-DNA insertion mutants of *ANAC044* and *ANAC085*.

DOI: https://doi.org/10.7554/eLife.43944.009

**Figure supplement 2.** *anac044* and *anac085* are tolerant to various DNA-damaging agents.

DOI: https://doi.org/10.7554/eLife.43944.010

**Figure supplement 2—source data 1.** Source data.

DOI: https://doi.org/10.7554/eLife.43944.011

**Figure supplement 3.** Root growth of *anac044-1*, *anac085-1* and *anac044-1 anac085-1* in the presence of DNA-damaging agents.

DOI: https://doi.org/10.7554/eLife.43944.012

(*Figure 3A and B*). This result suggests that *ANAC044* and *ANAC085* also participate in SOG1-mediated induction of stem cell death.

*Johnson et al. (2018)* investigated the recovery of *Arabidopsis* roots after an acute dose of ionizing radiation (150 Gy). They found that the stem cell niche was regenerated through experiencing cell death, and that root growth recovered after about one week. However, the *sog1* mutant underwent neither death nor regeneration of stem cells, leading to root growth arrest after long-term culture (*Johnson et al., 2018*). This observation suggests that SOG1-dependent programmed cell death triggers regeneration of the stem cell niche, and is thus essential for the recovery of root growth. As described above, *anac044-1 anac085-1* is also defective in DNA damage-induced stem cell death; therefore, we investigated root regeneration in *anac044-1 anac085-1* after a one-day treatment with bleomycin. In WT roots, PI-stained dead cells derived from stem cells and stele precursor cells gradually decreased and almost disappeared within five days (*Figure 3C*). In *sog1-101*, stem cell death did not occur, although we observed sporadic cell death across the meristematic zone and distorted tissue organization (*Figure 3C*), which were also described by *Johnson et al. (2018)*. On the other hand, *anac044-1 anac085-1* displayed no cell death in the stem cell niche or in the meristematic zone, and cells were properly arranged in the root tip throughout 7 days after recovery (*Figure 3C*). The *anac044-1 anac085-1 sog1-101* triple mutant showed disorganized root tissues as observed in *sog1-101* (*Figure 3C*), again indicating epistasis of *sog1-101* over *anac044-1 anac085-1*. These results suggest that stem cells in *sog1-101* and *anac044-1 anac085-1* have different characteristics; namely, DNA-damaged stem cells in *anac044-1 anac085-1*, but not *sog1-101*, can function in producing transit-amplifying cells, which constitute the root meristem. SOG1-dependent, but ANAC044/085-independent, pathway(s) are probably required for preserving the integrity of stem cells that have been exposed to DNA damage.

## ANAC044 and ANAC085 are not involved in homologous recombination-mediated DNA repair

In *Arabidopsis*, DNA repair and recombination are known to be induced through SOG1 in response to DSBs (*Yoshiyama et al., 2009*); indeed, our recent study showed that SOG1 directly controls several DNA repair-related genes, such as *RAD51* and *BRCA1* (*Ogita et al., 2018*). To investigate whether ANAC044 and ANAC085 participate in DNA repair, we examined the expression of DNA repair-related genes in the *anac044 anac085* double mutant. Five-day-old seedlings were treated with bleomycin for 10 hr, at which time *ANAC044* and *ANAC085* are highly induced in WT (*Figure 1—figure supplement 1A*), and total RNA was analysed using an Agilent Custom Microarray that covers all annotated *Arabidopsis* genes. Although the transcript levels of genes involved in non-homologous end-joining or mismatch excision repair did not change under our DNA damage conditions, several genes for homologous recombination (HR) (*BRCA1*, *RAD51A*, *RAD51B*, *RAD54L* and *MND1*) and nucleotide excision repair (*RPA1E*) were up-regulated in WT, but not in *sog1-101* (*Figure 4A*). In *anac044 anac085*, the up-regulated genes were normally induced by bleomycin treatment (*Figure 4A*), suggesting that neither ANAC044 nor ANAC085 is required for DSB-induced expression of DNA repair-related genes.

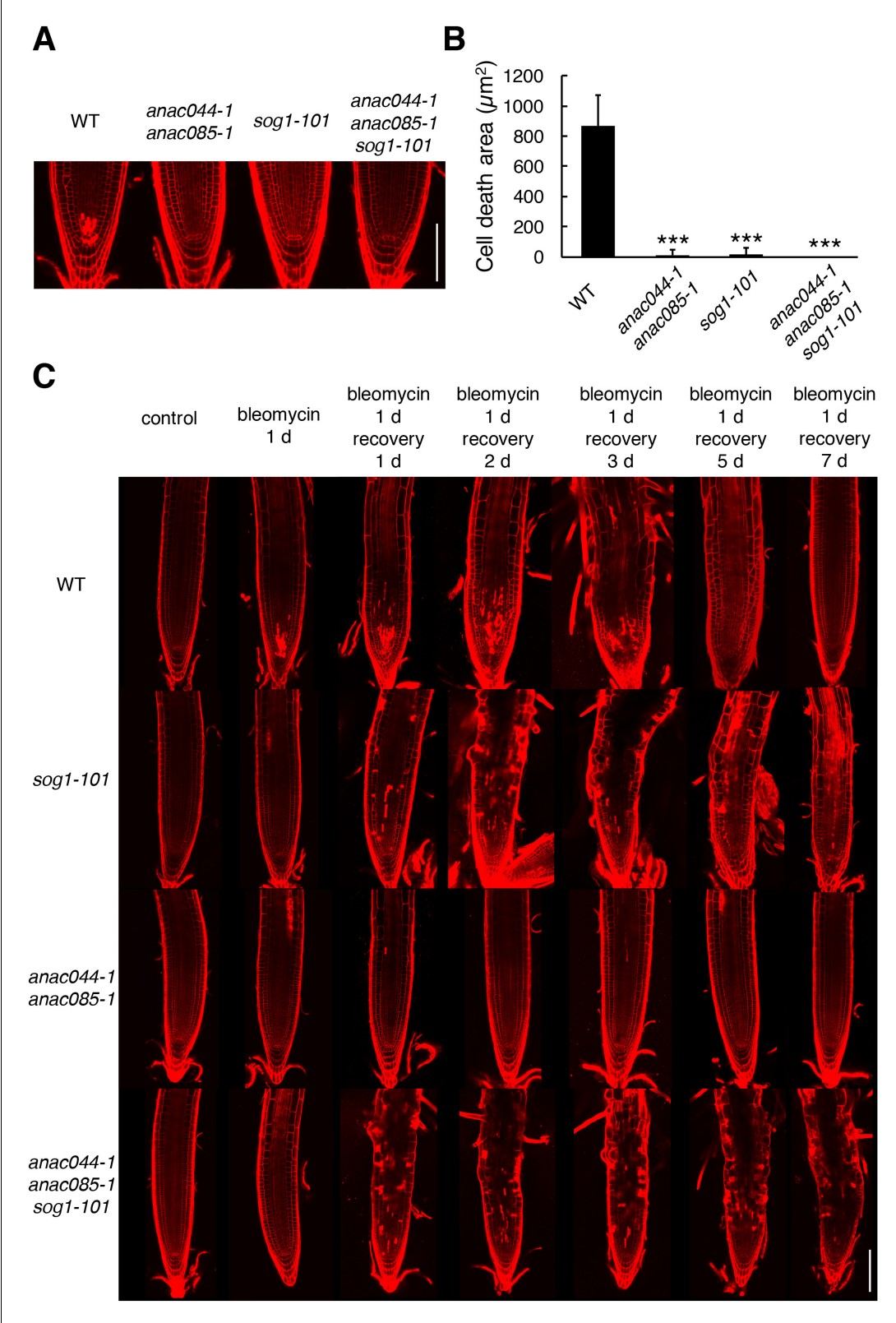

**Figure 3.** ANAC044 and ANAC085 are involved in DNA damage-induced stem cell death. (**A**) Root tips of WT, *anac044-1 anac085-1*, *sog1-101*, and *anac044-1 anac085-1 sog1-101*. Five-day-old seedlings were transferred to medium containing 0.6 µg/ml bleomycin and grown for 24 hr, followed by propidium iodide (PI) staining. Bar = 100 µm. (**B**) Cell death area in the root tip. The total area of cells stained with PI (**A**) was measured by ImageJ software. Data are presented as mean ± SD (n > 9). Significant differences from WT were determined by Student's *t*-test: ***, p<0.001. (**C**) Root tips of
*Figure 3 continued on next page*

*Figure 3 continued*

*sog1-101*, *anac044-1 anac085-1* and *anac044-1 anac085-1 sog1-101* after recovery from bleomycin-containing medium. Five-day-old seedlings were transferred to medium supplemented with 0.6 µg/ml bleomycin, and grown for 24 hr. The seedlings were then transferred back to medium without bleomycin, and grown for the indicated number of days. Root tips were observed after PI staining. Bar = 100 µm.

DOI: https://doi.org/10.7554/eLife.43944.014

The following source data is available for figure 3:

**Source data 1.** Source data.

DOI: https://doi.org/10.7554/eLife.43944.015

To test the functional relevance of ANAC044 and ANAC085 to DNA repair, we performed an assay to detect HR-mediated DNA repair events using the *beta-glucuronidase* (*GUS*) recombination reporter line (*Swoboda et al., 1994*). This assay uses parts of the *GUS* gene in directed orientation, which serve as a substrate for HR (*Figure 4B*). In cells where HR occurs, the *GUS* gene is restored, and GUS activity can be visualized and scored as blue spots after histochemical staining (*Swoboda et al., 1994*). Plants with the disrupted *GUS* gene were crossed with *sog1-101* and *anac044-1 anac085-1*, and blue spots were counted on leaves. Under normal growth conditions, *sog1-101* displayed fewer blue spots than WT (*Figure 4B*). Fifty grays of gamma irradiation increased the number of blue spots in both WT and *sog1-101*, although the number was significantly lower in *sog1-101* than in WT (*Figure 4B*). These results indicate that HR is suppressed in the *sog1-101* mutant. In contrast, no significant difference was observed in the number of blue spots between WT and *anac044-1 anac085-1* irrespective of gamma irradiation (*Figure 4B*), suggesting that ANAC044 and ANAC085 are not involved in HR.

## ANAC044 and ANAC085 inhibit cell cycle progression in response to DNA damage

The above results indicate that ANAC044 and ANAC085 are required for the part of the DDR that is essential for root growth retardation and stem cell death, but not for DNA repair. One possibility is that they are engaged in cell cycle arrest, because the *Arabidopsis myb3r3* and *myb3r5* mutants, which cannot induce G2 arrest upon exposure to DNA damage, also show defects in root growth retardation and stem cell death (*Chen et al., 2017*). To test this possibility, we first measured root meristem size by counting the number of cortical cells between the quiescent centre (QC) and the first elongated cell. Bleomycin treatment for 24 hr resulted in a 38% reduction in the meristem cell number in WT, while no significant reduction was observed in *anac044-1 anac085-1*, *sog1-101* or *anac044-1 anac085-1 sog1-101* (*Figure 5A*). This suggests that SOG1, ANAC044 and ANAC085 control cell division in the meristem and/or the transition from cell division to endoreplication under DNA damage conditions.

To clarify whether ANAC044 and ANAC085 are involved in DNA damage-induced G2 arrest, we conducted EdU incorporation experiments. Five-day-old roots were treated with or without bleomycin for 12 hr, and then incubated with EdU for 15 min. After EdU-labelled cells pass through the G2 phase, they are expected to show mitotic figures in the M phase (*Hayashi et al., 2013*); therefore, we counted the number of EdU-labelled cells with mitotic figures after 4 or 6 hr. In the absence of bleomycin, EdU-labelled cells with mitotic figures increased in WT, *anac044 anac085* and *sog1-101* after 4 hr (*Figure 5B*), implying that EdU-labelled cells entered the M phase. On the other hand, bleomycin treatment significantly suppressed the increase of EdU-positive cells with mitotic figures in WT, indicating an impairment in G2/M progression, whereas an increase was observed in *anac044 anac085* and *sog1-101* irrespective of bleomycin treatment (*Figure 5B*). This suggests that ANAC044 and ANAC085 as well as SOG1 are essential for inhibiting G2-to-M progression in response to DSBs.

## ANAC044 and ANAC085 control protein accumulation of the repressor-type MYB3R

We previously reported that when *Arabidopsis* cultured cells were treated with the DSB inducer zeocin, many G2/M-specific genes were suppressed, thereby causing G2 arrest and endoreplication (*Adachi et al., 2011*). To examine whether ANAC044 and ANAC085 are involved in repression of

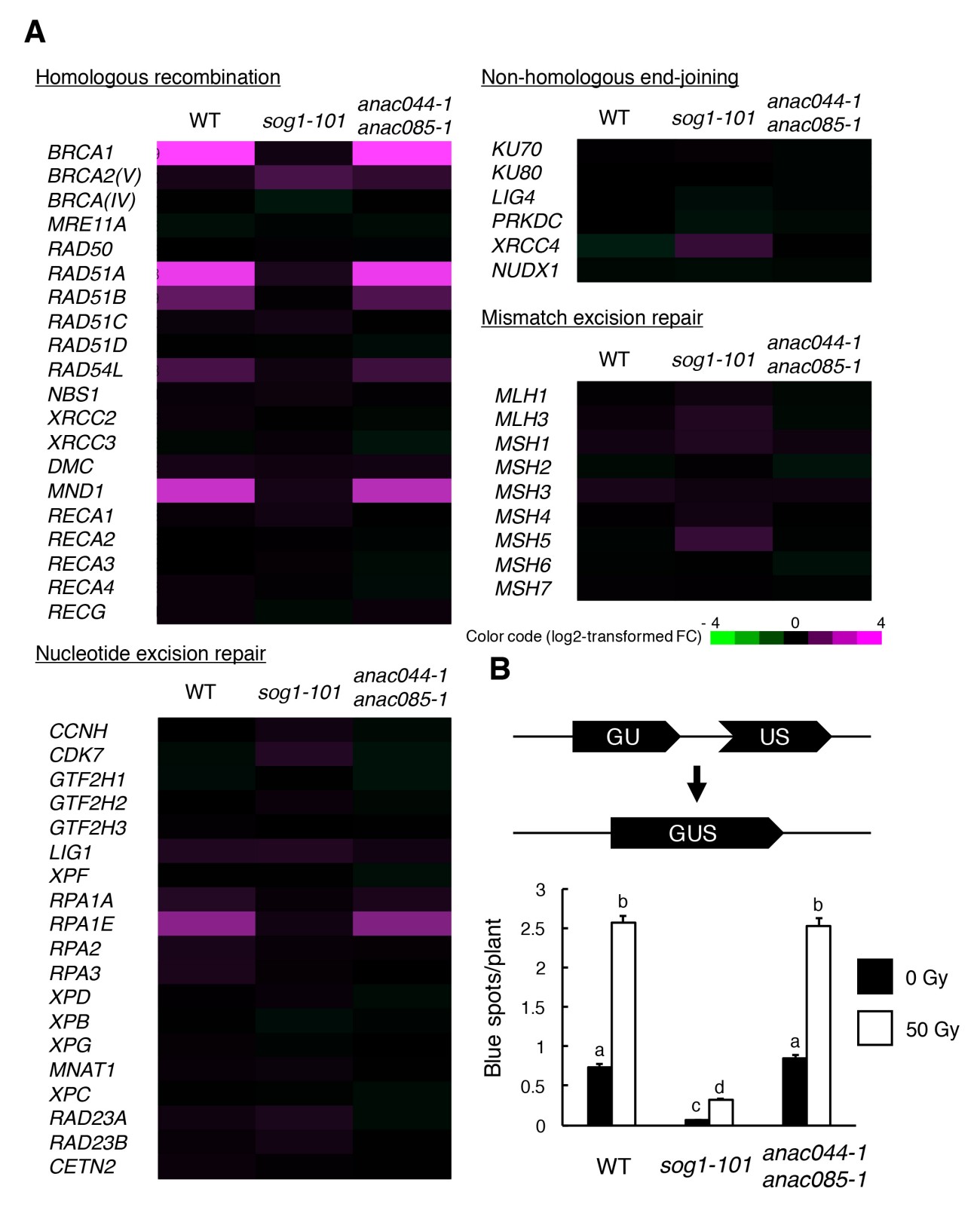

**Figure 4.** ANAC044 and ANAC085 are not required for HR-mediated DNA repair. (**A**) Transcriptional response of DNA repair-related genes to bleomycin. Five-day-old seedlings of WT, *sog1-101* and *anac044-1 anac085-1* were treated with or without 0.6 μg/ml bleomycin for 10 hr. Total RNA was extracted from root tips and subjected to microarray analysis. Purple and green colours indicate up- and down-regulation, respectively, of genes by bleomycin treatment. (**B**) HR assay. The *GUS* reporter constructs before and after HR are shown (upper panel). Two-week-old plants of WT, *sog1-101*

*Figure 4 continued on next page*

*Figure 4 continued*
and *anac044-1 anac085-1* carrying the *GUS* reporter construct were irradiated with or without gamma rays (50 Gy), and grown for 3 days. Numbers of blue spots on leaves were counted. Data are presented as mean ± SD (n = 50). Different letters indicate significant differences between samples (Student's *t*-test, p<0.05).
DOI: https://doi.org/10.7554/eLife.43944.016
The following source data is available for figure 4:

**Source data 1.** Source data.
DOI: https://doi.org/10.7554/eLife.43944.017

G2/M-specific genes, we measured the transcript levels of four MYB3R3 targets, *KNOLLE*, *CYCB1;2*, *EPS15 HOMOLOGY DOMAIN 2* (*EHD2*) and *PLEIADE/MAP65-3* (*Kobayashi et al., 2015*), in root tips. When WT seedlings were treated with bleomycin, the mRNA levels decreased over time and were reduced by half after 24 hr (*Figure 6A*). As reported previously, no reduction was observed in *sog1-101* (*Yoshiyama et al., 2009*), suggesting that SOG1 is essential to repress G2/M-specific genes. In the *anac044-1 anac085-1* double mutant, the transcript levels decreased until 12 hr as in WT; however, they did not further decrease beyond 24 hr of bleomycin treatment (*Figure 6A*). It is noteworthy that such an expression pattern is similar to that observed in *myb3r3* and *myb3r5* mutants, in which suppression of G2/M-specific genes is impaired at a later stage of the DDR (*Chen et al., 2017*).

It has been previously reported that the CDK inhibitors *SMR5* and *SMR7* are directly induced by SOG1 in response to DNA damage (*Yin et al., 2014*; *Ogita et al., 2018*). Some of the NAC-type transcription factors are known to form homo- or heterodimers to control their target genes (*Mitsuda et al., 2004*; *Yamaguchi et al., 2008*; *Gladman et al., 2016*). Therefore, we examined whether ANAC044 and ANAC085 also participate in controlling the expression of *SMR5* and *SMR7*. Our qRT-PCR showed that bleomycin treatment rapidly induced the expression of *SMR5* and *SMR7* in both WT and *anac044 anac085* (*Figure 6—figure supplement 1*), suggesting that ANAC044 or ANAC085 is not required for the immediate induction of CDK inhibitor genes. These data are consistent with the above-mentioned result that suppression of G2/M-specific genes was impaired only at a later stage of the DDR in *anac044 anac085* (*Figure 6A*).

The repressor-type MYB3R transcription factors MYB3R3 and MYB3R5 function in repressing G2/M-specific genes (*Kobayashi et al., 2015*). In the absence of DNA damage, they are actively degraded via the ubiquitin-proteasome pathway, while under DNA damage conditions, they accumulate to high levels and cause G2 arrest by inhibiting G2/M-specific gene expression (*Chen et al., 2017*). To examine whether ANAC044 and ANAC085 are associated with the protein stability of repressor-type MYB3Rs, we observed the accumulation in roots of the MYB3R3-GFP fusion protein, which was expressed under the 1.3 kb *MYB3R3* promoter. As reported previously, the GFP fluorescence was increased by bleomycin treatment in the root tip (*Chen et al., 2017*), whereas no increase was observed in *sog1-101* or *anac044-1 anac085-1* (*Figure 6B and C*). Immunoblot analysis using total protein from root tips also showed bleomycin-induced accumulation of MYB3R3-GFP in WT, but not in *anac044-1 anac085-1* (*Figure 6D*). Note that the *MYB3R3* transcript level was unaltered by mutations in *SOG1*, *ANAC044* or *ANAC085* (*Figure 6—figure supplement 1*). These results suggest that ANAC044 and ANAC085 as well as SOG1 are involved in DNA damage-induced accumulation of the repressor-type MYB3Rs. We have reported that expression of the activator-type *MYB3R4* decreases in response to DNA damage at the mRNA level, and that this down-regulation requires SOG1 (*Chen et al., 2017*). Our qRT-PCR showed that *MYB3R4* transcripts were reduced in *anac044 anac085* in a similar manner to that in WT (*Figure 6—figure supplement 1*), suggesting that ANAC044 and ANAC085 function in controlling the level of repressor-type, but not of activator-type, MYB3Rs.

Our measurement of root growth showed that *anac044 anac085* was as tolerant to bleomycin treatment as *myb3r3-1*, a knockout mutant of *MYB3R3* (*Chen et al., 2017*) (*Figure 6—figure supplement 2*). Moreover, the *anac044 anac085 myb3r3-1* triple mutant also exhibited the same sensitivity to bleomycin (*Figure 6—figure supplement 2*), indicating that ANAC044/085 and MYB3R3 function in the same pathway. Taken together, our results suggest that *ANAC044* and *ANAC085*, which are

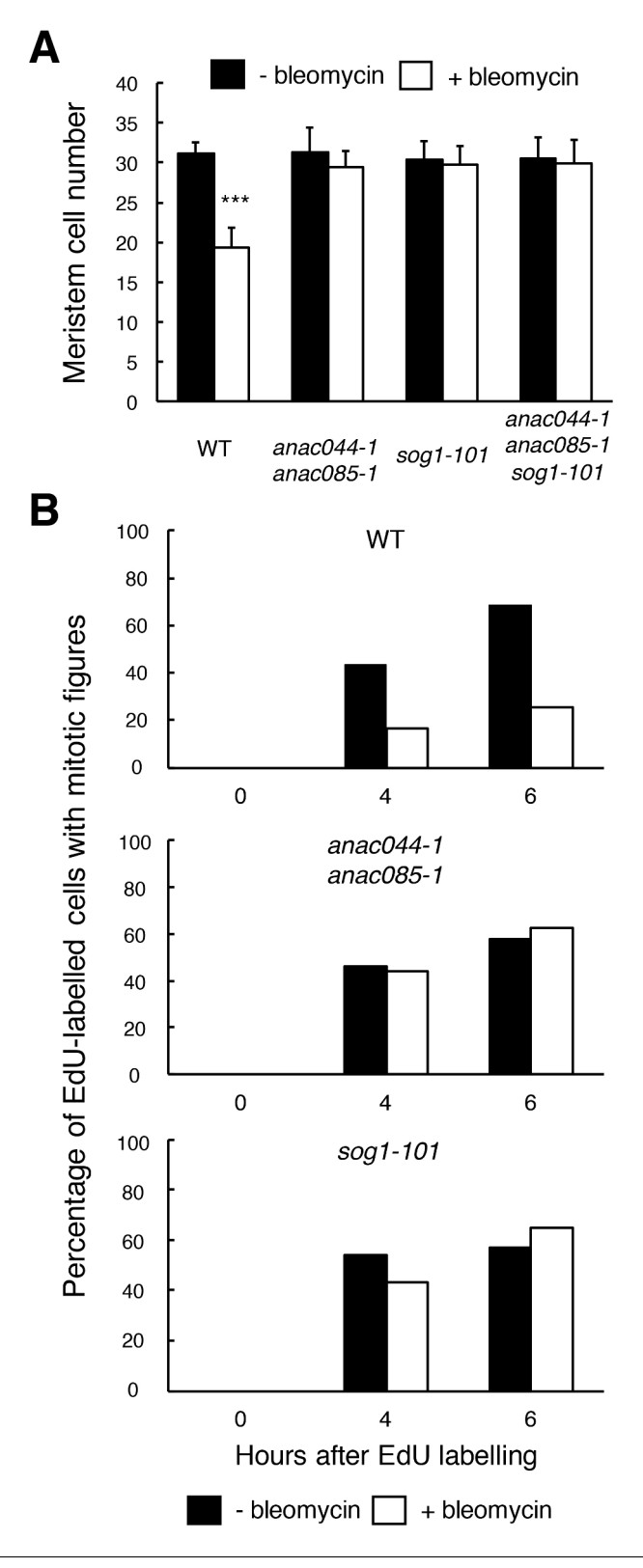

**Figure 5.** ANAC044 and ANAC085 are required for DNA damage-induced cell cycle arrest. (**A**) Cell number in the meristematic zone of WT, *anac044-1 anac085-1*, *sog1-101* and *anac044-1 anac085-1 sog1-101*. Five-day-old seedlings were transferred to MS medium supplemented with or without 0.6 μg/ml bleomycin, and the number of cortex cells between the QC and the first elongated cell was counted after 24 hr. Data are presented as

*Figure 5 continued on next page*

*Figure 5 continued*

mean ± SD (n = 20). Significant differences from the non-treated control were determined by Student's *t*-test: ***, p<0.001. (B) Cell cycle progression through the G2/M phase. Five-day-old seedlings of WT, *anac044-1 anac085-1* and *sog1-101* were transferred to MS medium supplemented with or without 0.6 µg/ml bleomycin for 12 hr, and pulse-labelled with EdU for 15 min. Seedlings were then transferred back to MS medium with or without bleomycin, and collected at the indicated time points. Root meristematic cells were double-stained with EdU and DAPI, and cells with mitotic figures were counted. Data are presented as the percentage of EdU-labelled cells among those with mitotic figures (n > 20).
DOI: https://doi.org/10.7554/eLife.43944.018
The following source data is available for figure 5:

**Source data 1.** Source data.
DOI: https://doi.org/10.7554/eLife.43944.019

induced by DNA damage-activated SOG1, enhance protein accumulation of repressor-type MYB3Rs, thereby suppressing G2/M-specific genes and causing G2 arrest.

## ANAC044 overexpression inhibits G2/M progression under DNA damage conditions

To further examine the role of the ANAC044/085-mediated pathway in cell cycle arrest, we generated transgenic plants overexpressing *ANAC044* fused to the glucocorticoid receptor, which is activated by dexamethasone (Dex). We used two lines, of which #1 showed a higher expression of *ANAC044-GR* than #2, for further analyses (*Figure 7A*). Five-day-old seedlings grown on MS medium were transferred onto medium supplemented with or without 10 µM Dex and/or 0.6 µg/ml bleomycin, and root length was measured for 5 days. In the absence of bleomycin, the overexpression lines displayed similar root growth to WT irrespective of Dex treatment (*Figure 7B*), suggesting that *ANAC044* overexpression by itself does not inhibit cell division. However, in the presence of bleomycin, root growth was further retarded in the overexpression lines; #1 showed much slower root growth than #2, which has lower *ANAC044-GR* expression (*Figure 7A and B*; *Figure 7—figure supplement 1*). These results clearly suggest that *ANAC044* overexpression inhibits root growth under DNA damage conditions.

Next, we conducted EdU incorporation experiments. In the absence of bleomycin, the number of EdU-positive cells with mitotic figures increased in both WT and the overexpression line #1 (*Figure 7C*). However, in the presence of bleomycin, the increase of EdU-positive cells with mitotic figures was inhibited in WT, and more dramatically so in #1 (*Figure 7C*). The expression of G2/M-specific genes was also severely suppressed in #1 compared to WT after bleomycin treatment (*Figure 7D*). These results suggest that DSB-induced inhibition of G2/M progression is further enhanced by *ANAC044* overexpression.

## ANAC044/085 and the repressor-type MYB3R are involved in heat stress-induced G2 arrest

The above results indicate that the ANAC044/085-mediated pathway plays a major role in DNA stress-induced cell cycle arrest. To examine whether the same signalling module also functions in other stress responses, we investigated the involvement of ANAC044 and ANAC085 in the response to heat and osmotic stresses, which are known to retard G2 and G1 phase progression, respectively, in maize roots (*Zhao et al., 2014*). When transgenic *Arabidopsis* seedlings harbouring the cell cycle marker system Cytrap (*Yin et al., 2014*) were grown at higher temperature (37°C) for 24 hr, root cells expressing the late G2/M marker *CYCB1;1* increased significantly, while those expressing the S/G2 marker *CDT1a* were dramatically reduced (*Figure 8A and B*). On the other hand, when the Cytrap line was treated with 400 mM mannitol for 24 hr, S/G2 phase cells decreased by half, and G1 phase cells, which are represented by the lack of expression of both *CYCB1;1* and *CDT1a* markers, significantly increased (*Figure 8—figure supplement 1A and B*). These results are consistent with the previous observation in maize (*Zhao et al., 2014*). We found that the *anac044 anac085* mutant exhibited the same sensitivity as WT to mannitol (*Figure 8—figure supplement 1C*), and that the promoter activity of *ANAC044* was not elevated by mannitol treatment (*Figure 8—figure*

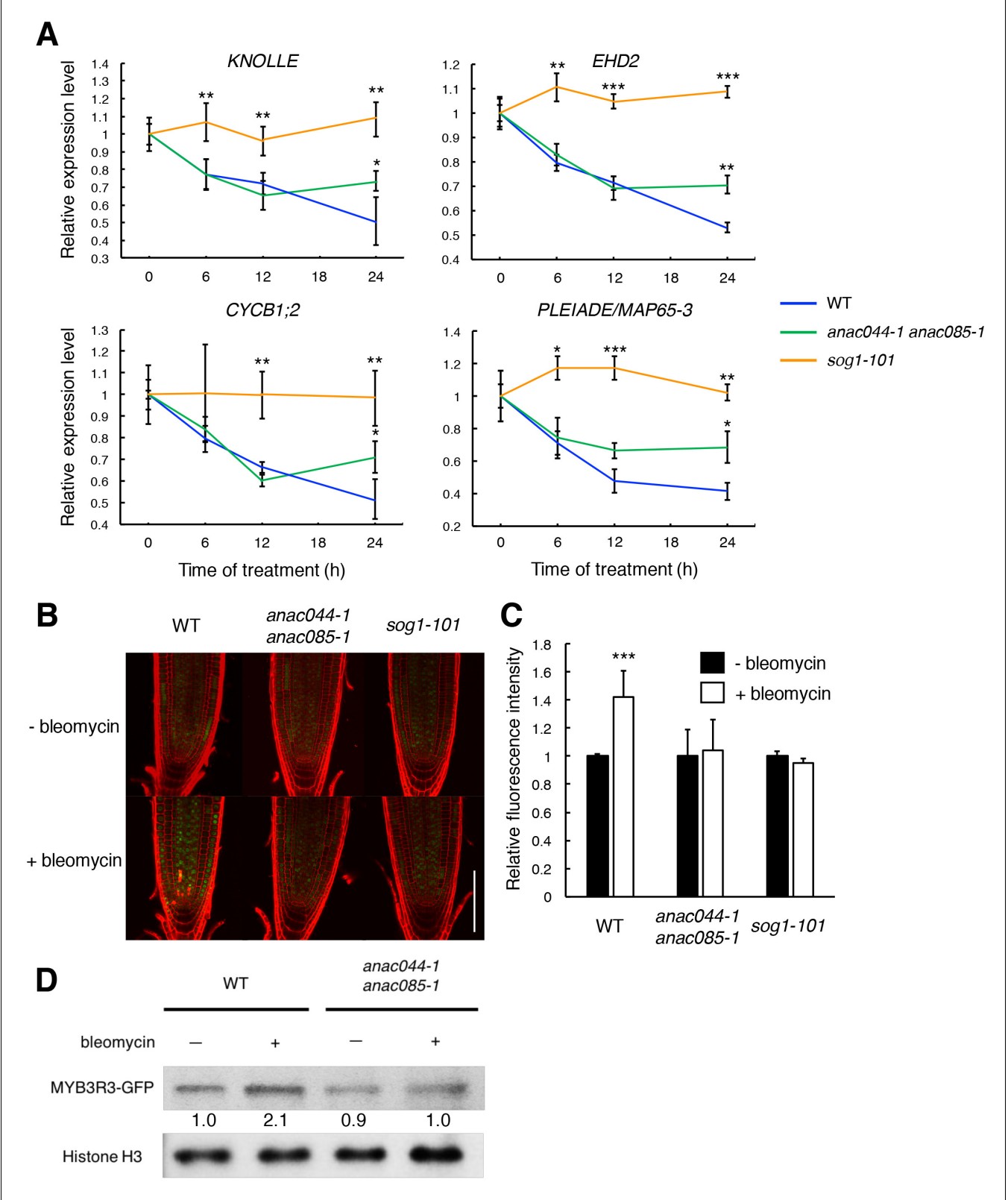

**Figure 6.** ANAC044 and ANAC085 promote Rep-MYB accumulation in response to DNA damage. (**A**) Transcript levels of *KNOLLE*, *CYCB1;2*, *EHD2* and *PLEIADE/MAP65-3* after bleomycin treatment. Five-day-old seedlings of WT, *anac044-1 anac085-1* and *sog1-101* were transferred to MS medium containing 0.6 µg/ml bleomycin, and grown for 0, 6, 12 and 24 hr. Total RNA was extracted from root tips. mRNA levels were normalized to that of *ACTIN2*, and are indicated as relative values, with that for 0 hr set to 1. Data are presented as mean ± SD (n = 3). Significant differences from WT were
*Figure 6 continued on next page*

*Figure 6 continued*

determined by Student's *t*-test: *, p<0.05; **, p<0.01; ***, p<0.001. (**B–D**) MYB3R3 accumulation in the root tip. Five-day-old seedlings of WT, *anac044-1 anac085-1* and *sog1-101* harbouring *ProMYB3R3:MYB3R3-GFP* were transferred to MS medium supplemented with or without 0.6 µg/ml bleomycin, and grown for 24 hr. GFP fluorescence was observed after counterstaining with PI (**B**). Bar = 100 µm. Intensities of GFP fluorescence in the root tip are shown as relative values, with that of non-treated control set to 1 (**C**). Data are presented as mean ± SD (n = 5). Significant differences from the non-treated control were determined by Student's *t*-test: ***, p<0.001. Twenty micrograms of total protein extracted from root tips were subjected to immunoblotting using antibodies against GFP or histone H3 (**D**). Relative levels of MYB3R3-GFP are expressed as fold change, normalized with respect to the corresponding band of histone H3.

DOI: https://doi.org/10.7554/eLife.43944.020

The following source data and figure supplements are available for figure 6:

**Source data 1.** Source data.
DOI: https://doi.org/10.7554/eLife.43944.025
**Source data 2.** Uncut blot.
DOI: https://doi.org/10.7554/eLife.43944.026
**Figure supplement 1.** mRNA levels of *SMR5*, *SMR7*, *MYB3R3* and *MYB3R4* after bleomycin treatment.
DOI: https://doi.org/10.7554/eLife.43944.021
**Figure supplement 1—source data 1.** Source data.
DOI: https://doi.org/10.7554/eLife.43944.022
**Figure supplement 2.** Root growth of *anac044-1 anac085-1*, *myb3r3-1* and *anac044-1 anac085-1 myb3r3-1* in the presence of bleomycin.
DOI: https://doi.org/10.7554/eLife.43944.023
**Figure supplement 2—source data 1.** Source data.
DOI: https://doi.org/10.7554/eLife.43944.024

*supplement 1D*). This suggests that ANAC044 and ANAC085 are not associated with the osmotic stress-induced inhibition of G1 progression; therefore, we focus on heat stress hereafter.

*anac044 anac085* seedlings grown on MS medium at 22°C for 5 days were incubated at 37°C for 24 hr, and root length was measured after transfer to 22°C. As shown in *Figure 8C*, root growth inhibition by heat stress was less evident in *anac044 anac085* than in WT, indicating heat stress tolerance (*Figure 8—figure supplement 2*). A tolerance to heat stress was also observed in a reduction of cell number in the root meristem (*Figure 8D*). *sog1-101* exhibited the same sensitivity as WT (*Figure 8C*; *Figure 8—figure supplement 2*), suggesting that *ANAC044* and *ANAC085* inhibit root growth independently of SOG1. We next examined the transcriptional response of *ANAC044* and *ANAC085* to heat treatment. qRT-PCR showed that the transcript levels of *ANAC044* and *ANAC085* were highly elevated by heat stress in WT, and to the same extent in *sog1-101* (*Figure 8E*). A similar result was also obtained using the *ProANAC044:GUS* reporter line (*Figure 8F*). These results suggest that *ANAC044* and *ANAC085* perceive heat stress signals independently of the SOG1-mediated pathway, and that their induction leads to G2 arrest.

The next question is whether ANAC044 and ANAC085 inhibit G2 phase progression by controlling protein accumulation of repressor-type MYB3Rs, as in the case of DNA stress. In the *ProMYB3R3:MYB3R3-GFP* reporter line, the GFP fluorescence was increased by incubation of seedlings at 37°C for 24 hr, but no such induction was observed in *anac044 anac085* (*Figure 9A and B*). This result was supported by immunoblotting (*Figure 9C*). The *MYB3R3* transcript level was elevated by heat treatment in both WT and *anac044 anac085* (*Figure 9—figure supplement 1*), indicating that transcriptional induction does not affect protein abundance; rather, protein-level regulation is crucial in controlling MYB3R3 accumulation. Measurement of root growth and counting the meristematic cell number showed that *myb3r3-1* and the *anac044 anac085 myb3r3-1* triple mutant exhibited the same tolerance to heat stress as *anac044 anac085* (*Figure 8D*; *Figure 9D*; *Figure 9—figure supplement 2*). These results suggest that ANAC044 and ANAC085 are required for MYB3R3 accumulation in response to heat stress, and that the ANAC044/085–MYB3R3/5 signalling pathway is a central module controlling G2 arrest in stress responses.

## Discussion

In this study, we revealed that DNA damage-activated SOG1 causes cell cycle arrest through two other NAC-type transcription factors, ANAC044 and ANAC085, which promote the accumulation of

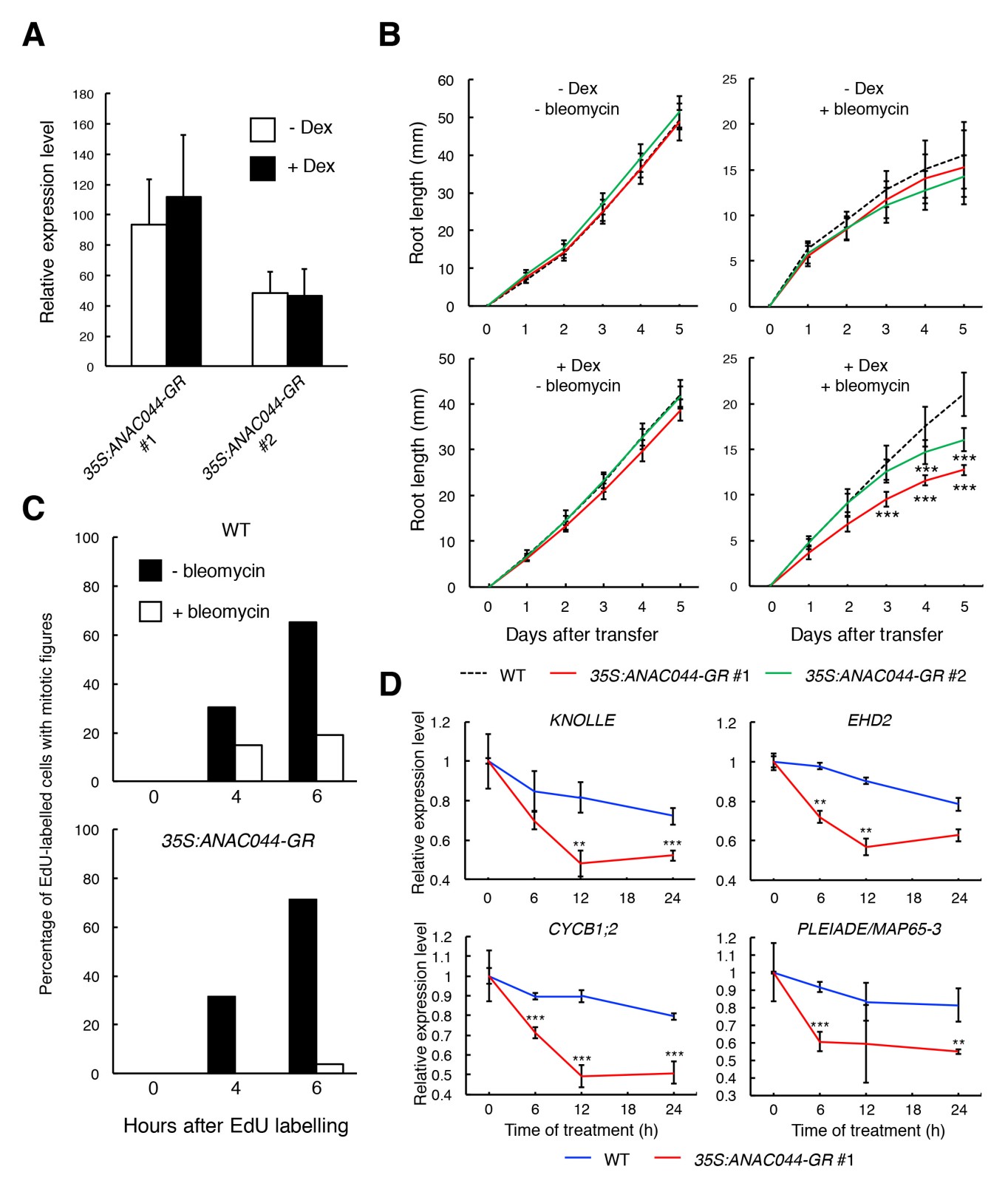

**Figure 7.** *ANAC044*-overexpressing plants exhibit hypersensitivity to DNA damage. (**A**) Transcript levels of *ANAC044-GR*. Five-day-old seedlings of two independent lines of *35S:ANAC044-GR* (#1 and #2) were treated with or without 10 μM Dex for 24 hr. mRNA levels were quantified by qRT-PCR using *ANAC044* primers and normalized to that of *ACTIN2*, and are indicated as relative values, with that of endogenous *ANAC044* in WT set to 1. Data are presented as mean ± SD (n = 3). (**B**) Root growth of *35S:ANAC044-GR* seedlings. Five-day-old seedlings were transferred to MS medium supplemented

*Figure 7 continued on next page*

*Figure 7 continued*
with or without 10 μM Dex and/or 0.6 μg/ml bleomycin, and root length was measured every 24 hr. Data are presented as mean ± SD (n > 12). Significant differences from WT were determined by Student's *t*-test: ***, p<0.001. (C) Cell cycle progression through the G2/M phase. Five-day-old seedlings of WT and *35S:ANAC044-GR* #1 were grown in 10 μM Dex-containing medium with or without 0.6 μg/ml bleomycin for 12 hr. After pulse-labelling with EdU for 15 min, seedlings were transferred back to Dex-containing medium with or without bleomycin, and collected after 0, 4 and 6 hr. Root meristematic cells were double-stained with EdU and DAPI, and cells with mitotic figures were counted. Data are presented as the percentage of EdU-labelled cells among those with mitotic figures (n > 20). (D) Transcript levels of G2/M-specific genes. Five-day-old seedlings of WT and *35S: ANAC044-GR* #1 were transferred to MS medium containing 10 μM Dex and 0.6 μg/ml bleomycin, and grown for 0, 6, 12 and 24 hr. Total RNA was extracted from root tips and subjected to qRT-PCR. mRNA levels were normalized to that of *ACTIN2*, and are indicated as relative values, with that for 0 hr set to 1. Data are presented as mean ± SD (n = 3). Significant differences from WT were determined by Student's *t*-test: **, p<0.01; ***, p<0.001.
DOI: https://doi.org/10.7554/eLife.43944.027
The following source data and figure supplement are available for figure 7:

**Source data 1.** Source data.
DOI: https://doi.org/10.7554/eLife.43944.029
**Figure supplement 1.** Root growth of *35S:ANAC044-GR* seedlings.
DOI: https://doi.org/10.7554/eLife.43944.028

Rep-MYB proteins and suppress G2/M-specific genes, thereby inducing G2 arrest. We also tested heat and osmotic stresses that retard cell cycle progression in G2 and G1 phases, respectively. Although the *anac044 anac085* mutant was as sensitive as WT to osmotic stress, it was tolerant to heat stress and defective in heat-induced accumulation of Rep-MYB. These data suggest that ANAC044 and ANAC085 play a crucial role in controlling cell cycle arrest at G2, but not at the other phases, in response to stresses (*Figure 10*). Deployment of the ANAC044/085-mediated signalling module indicates that although eukaryotic cells commit to enter a new cell cycle during G1 (*Bertoli et al., 2013*), plants have an additional critical point at G2 for deciding whether or not cells resume proliferation after escaping from stress. Indeed, DNA damage enhances the onset of endor-eplication by inhibiting G2/M progression, thereby prohibiting resumption of the mitotic cell division (*Adachi et al., 2011*). The deployment of such a module controlling G2 progression may be specific to plants, in which the induction of endoreplication is a sensible strategy to avoid cell death and enable continuous growth under stressful conditions (*Adachi et al., 2011*; *De Veylder et al., 2011*; *Radziejwoski et al., 2011*). Since ANAC044 and ANAC085 are also involved in DNA damage-induced stem cell death (*Figure 3A*), G2 arrest may be prerequisite for choosing between life and death of mitotic cells as well. Further studies will reveal how the ANAC044/085-mediated pathway is associated with the onset of endoreplication and cell death in response to stresses.

*Bourbousse et al. (2018)* conducted transcriptomic analyses over a 24 hr time course after gamma irradiation and revealed that Rep-MYBs play a major role in repressing cell cycle-related genes in response to DSBs. We previously reported that Rep-MYBs are actively degraded through the ubiquitin-proteasome pathway, and that this degradation is triggered by CDK phosphorylation (*Chen et al., 2017*). Under normal growth conditions, higher CDK activity enhances Rep-MYB degradation and maintains the expression of G2/M-specific genes; under DNA damage conditions, CDK activity is reduced by the induction and repression, respectively, of genes for CDK inhibitors and Act-MYB, resulting in stabilization of Rep-MYBs and suppression of G2/M-specific genes (*Chen et al., 2017*). In the *anac044 anac085* mutant, DNA damage up- and down-regulates the expression of CDK inhibitors (*SMR5* and *SMR7*) and *MYB3R4*, respectively, to the same extent as in WT (*Figure 6—figure supplement 1*), suggesting that the initial reduction of CDK activities occurs normally. This idea is supported by the fact that transcript levels of G2/M-specific genes, which are induced by MYB3R4 (*Haga et al., 2011*), decreased by 12 hr after bleomycin treatment as observed in WT (*Figure 6A*). However, in *anac044 anac085*, further reduction in G2/M-specific gene expression after 12 hr was suppressed (*Figure 6A*), indicating that ANAC044 and ANAC085 stabilize Rep-MYBs at a later stage of the DDR by controlling factors other than CDK activity. One possibility is that ANAC044 and ANAC085 inhibit the expression, accumulation or activity of the degradation machinery for Rep-MYBs. *Chen et al. (2017)* and *Kobayashi et al. (2015)* reported that under normal growth conditions, MYB3R3 protein preferentially accumulates around early S phase, and that Rep-MYBs repress transcription of target genes along the cell cycle except for G2/M. Therefore, it is likely that ubiquitin E3 ligases such as SCF (Skp-Cullin1-F-box) mark Rep-MYBs by polyubiquitination

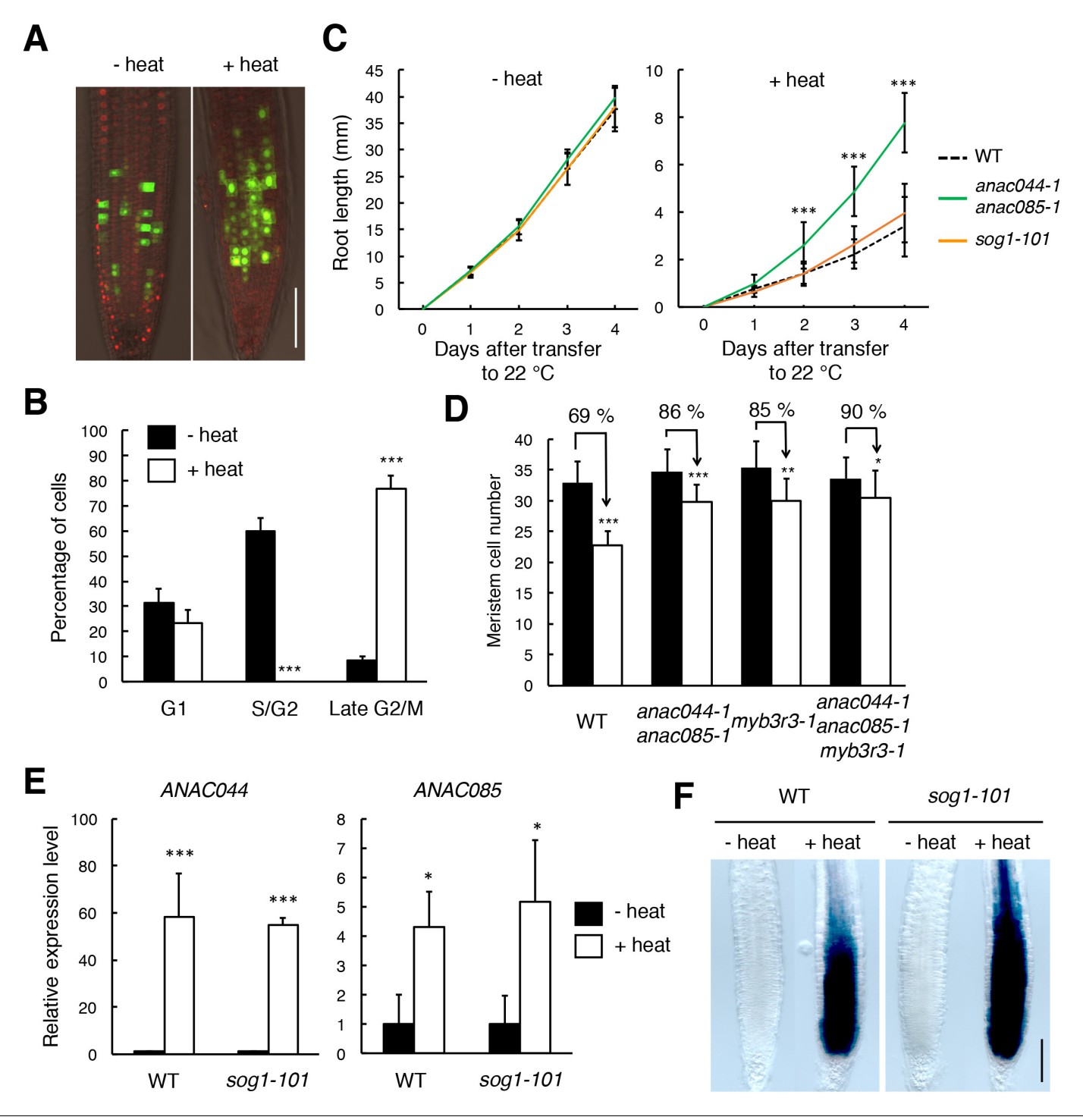

**Figure 8.** ANAC044 and ANAC085 are involved in the heat stress response in roots. (**A, B**) Heat stress-induced cell cycle arrest in the root tip. Five-day-old seedlings carrying the cell cycle marker system Cytrap were incubated at 22°C (- heat) or 37°C (+heat) for 24 hr. Signals of *ProHTR2:CDT1a (C3)-RFP* (S/G2 marker; red) and *ProCYCB1;1:CYCB1;1-GFP* (late G2/M marker; green) were observed (**A**). Bar = 100 µm. Cells with RFP or GFP signals were estimated to be in S/G2 or late G2/M, respectively, and those with no fluorescence were counted as G1-phase cells (**B**). Data are presented as mean ± SD (n = 10). Significant differences from the control (22°C) were determined by Student's *t*-test: ***, p<0.001. (**C**) Root growth of WT, *anac044-1 anac085-1* and *sog1-101* under heat stress. Five-day-old seedlings were incubated at 22°C (- heat) or 37°C (+heat) for 24 hr, and transferred to 22°C to measure root length every 24 hr. Data are presented as mean ± SD (n > 13). Significant differences from WT were determined by Student's *t*-test: ***, p<0.001. (**D**) Cortical cell number in the root meristem after heat stress. Five-day-old seedlings of WT, *anac044-1 anac085-1*, *myb3r3-1* and *anac044-1*

*Figure 8 continued on next page*

*Figure 8 continued*

*anac085-1 myb3r3-1* were incubated at 22°C (black bars) or 37°C (white bars) for 6 hr. Data are presented as mean ± SD (n > 15). Significant differences from the control (22°C) were determined by Student's *t*-test: *, p<0.05; **, p<0.01; ***, p<0.001. (E) Transcript levels of *ANAC044* and *ANAC085* under heat stress. Five-day-old seedlings of WT and *sog1-101* were incubated at 22°C (- heat) or 37°C (+heat) for 24 hr, and total RNA was extracted. mRNA levels were normalized to that of *ACTIN2*, and are indicated as relative values, with that for the control (22°C) set to 1. Data are presented as mean ± SD (n = 3). Significant differences from the control (22°C) were determined by Student's *t*-test: *, p<0.05, ***, p<0.001. (F) GUS staining of WT and *sog1-101* roots harbouring *ProANAC044:GUS*. Five-day-old seedlings were incubated at 22°C (- heat) or 37°C (+heat) for 24 hr, followed by GUS staining. Bar = 100 µm.

DOI: https://doi.org/10.7554/eLife.43944.030

The following source data and figure supplements are available for figure 8:

**Source data 1.** Source data.
DOI: https://doi.org/10.7554/eLife.43944.034
**Figure supplement 1.** ANAC044 and ANAC085 are not involved in the osmotic stress response.
DOI: https://doi.org/10.7554/eLife.43944.031
**Figure supplement 1—source data 1.** Source data.
DOI: https://doi.org/10.7554/eLife.43944.032
**Figure supplement 2.** Root growth of *anac044-1 anac085-1* and *sog1-101* after heat stress.
DOI: https://doi.org/10.7554/eLife.43944.033

for proteolysis during the cell cycle. A previous study showed that transcript levels of several F-box genes were reduced by gamma-irradiation (*Culligan et al., 2006*); therefore, such factors related to ubiquitin-mediated proteolysis may be under the control of ANAC044 and ANAC085, thereby regulating the protein stability of Rep-MYBs under DNA damage conditions.

We revealed that ANAC044 and ANAC085 are not required for induction of DNA repair-related genes, and that the frequency of HR was dramatically reduced in *sog1-101*, but not in *anac044 anac085* (*Figure 4*). This indicates that ANAC044 and ANAC085 are engaged in inhibiting cell division, whereas SOG1 is involved in DNA repair as well as cell cycle regulation (*Ogita et al., 2018*). *Johnson et al. (2018)* reported that when the *sog1* mutant is exposed to an acute dose of ionizing radiation, SOG1-independent cell death occurs in the root meristem, and root organization is severely impaired. We also observed a similar phenotype by transient treatment with bleomycin. It is noteworthy that unlike *sog1*, *anac044 anac085* displayed proper arrangement of root cells after recovery from bleomycin treatment (*Figure 3C*). Therefore, although stem cell death is suppressed in both *sog1* and *anac044 anac085*, stem cells in *sog1*, but not *anac044 anac085*, may seriously suffer from DNA damage, thus failing to undergo mitotic cell division and to maintain root organization. It is likely that defects in activation of DNA repair make stem cells hypersensitive to DNA damage in the *sog1* mutant. On the other hand, the phenotype of *anac044 anac085* suggests that SOG1-mediated activation of DNA repair is sufficient to preserve stem cells, whereas stem cell death and regeneration of new stem cells are probably also important to preserve genome integrity in stem cells.

In *Arabidopsis*, SOG1 is phosphorylated by ATM and ATR; ATM phosphorylates five serine-glutamine (SQ) motifs in the C-terminal region of SOG1 upon exposure to DSBs (*Yoshiyama et al., 2013*; *Sjogren et al., 2015*; *Yoshiyama et al., 2017*). Such phosphorylation is essential for SOG1's ability to bind to target promoters (*Ogita et al., 2018*). In contrast, ANAC044 and ANAC085 lack SQ motifs, raising the possibility that they do not require phosphorylation of the SQ motif for their transcriptional function. Their involvement in cell cycle arrest in response to heat stress also indicates that ATM/ATR-mediated phosphorylation is not necessary for activation. Nonetheless, we found that *ANAC044* overexpression by itself could not induce cell cycle arrest, and that DNA damage signals are required for ANAC044 function (*Figure 7*). This suggests that ANAC044/085 are phosphorylated by other kinase(s) or subjected to other modifications that are responsive to stresses. The DNA damage sensitivity of the *anac044 anac085* double mutant was indistinguishable from that of the single mutants, implying that ANAC044 and ANAC085 work together in DDRs. Several NAC transcription factors are known to form homo- or heterodimers (*Mitsuda et al., 2004*; *Yamaguchi et al., 2008*; *Gladman et al., 2016*), suggesting the possibility that ANAC044 and ANAC085 form dimers to act on target genes. This hypothesis is supported by our obsevation that *ANAC044* overexpression further retarded G2 progression only in the presence of bleomycin, which induces *ANAC085*. On the

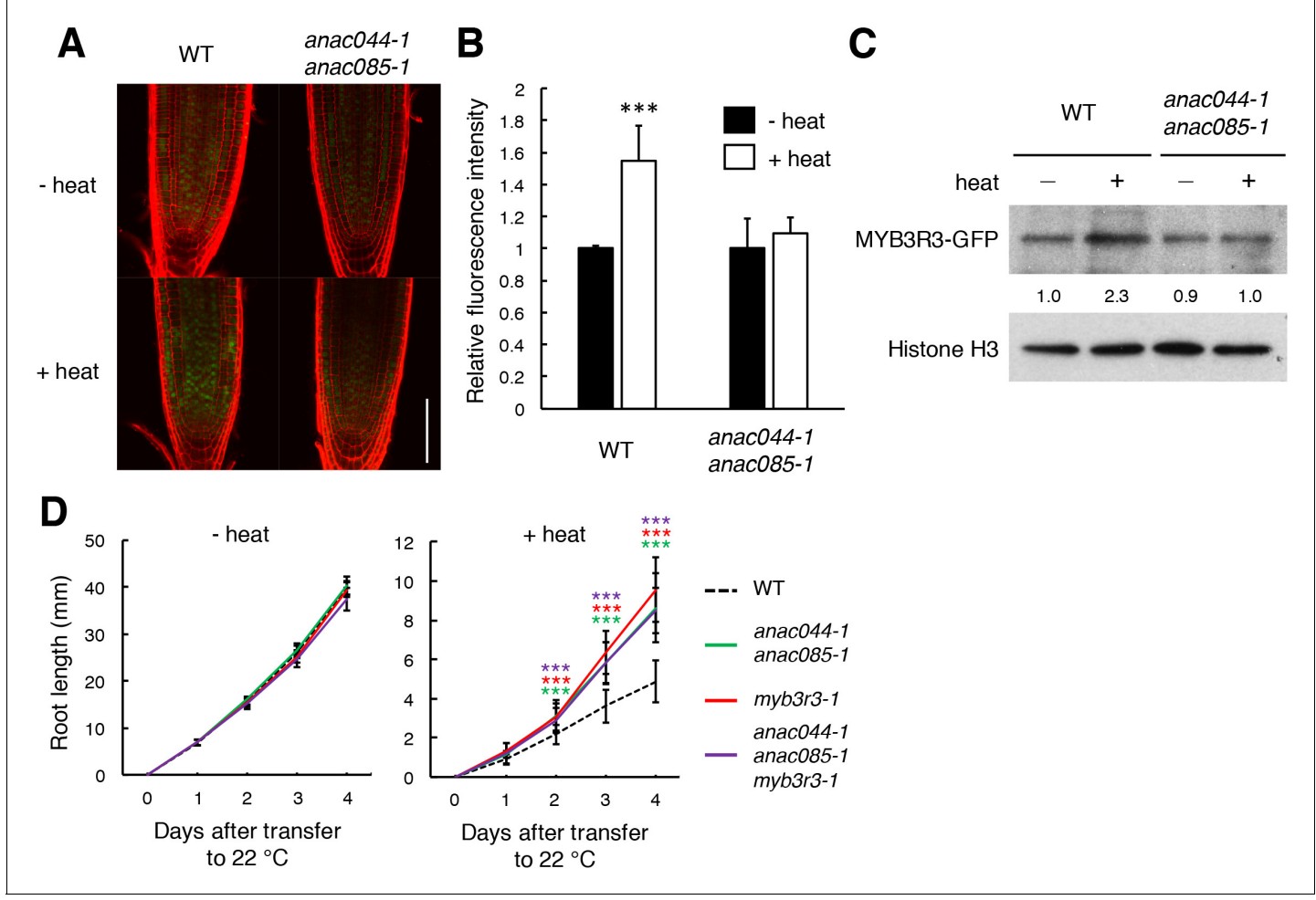

**Figure 9.** Heat stress induces ANAC044/085-mediated Rep-MYB accumulation in the root tip. (**A–C**) Rep-MYB accumulation under heat stress. Five-day-old seedlings of WT and *anac044-1 anac085-1* harbouring *ProMYB3R3:MYB3R3-GFP* were incubated at 22°C (- heat) or 37°C (+heat) for 12 hr. GFP fluorescence was observed after counterstaining with PI (**A**). Bar = 100 μm. Fluorescence intensities are indicated as relative values, with that of the control (22°C) set to 1 (**B**). Data are presented as mean ± SD (n > 5). Significant differences from the control (22°C) were determined by Student's *t*-test: ***, p<0.001. Twenty micrograms of total protein were extracted from root tips and subjected to immunoblotting using antibodies against GFP or histone H3 (**C**). Relative levels of MYB3R3-GFP are expressed as fold change, normalized with respect to the corresponding band of histone H3. (**D**) Root growth of WT, *anac044-1 anac085-1*, *myb3r3-1* and *anac044-1 anac085-1 myb3r3-1* under heat stress. Five-day-old seedlings were incubated at 22°C (- heat) or 37°C (+heat) for 24 hr, and transferred to 22°C to measure root length every 24 hr. Data are presented as mean ± SD (n > 12). Significant differences from WT were determined by Student's *t*-test: ***, p<0.001.
DOI: https://doi.org/10.7554/eLife.43944.035

The following source data and figure supplements are available for figure 9:

**Source data 1.** Source data.
DOI: https://doi.org/10.7554/eLife.43944.039
**Source data 2.** Uncut blot.
DOI: https://doi.org/10.7554/eLife.43944.040
**Figure supplement 1.** *MYB3R3* expression under heat stress.
DOI: https://doi.org/10.7554/eLife.43944.036
**Figure supplement 1—source data 1.** Source data.
DOI: https://doi.org/10.7554/eLife.43944.037
**Figure supplement 2.** Root growth of *anac044-1 anac085-1*, *myb3r3-1* and *anac044-1 anac085-1 myb3r3-1* after heat stress.
DOI: https://doi.org/10.7554/eLife.43944.038

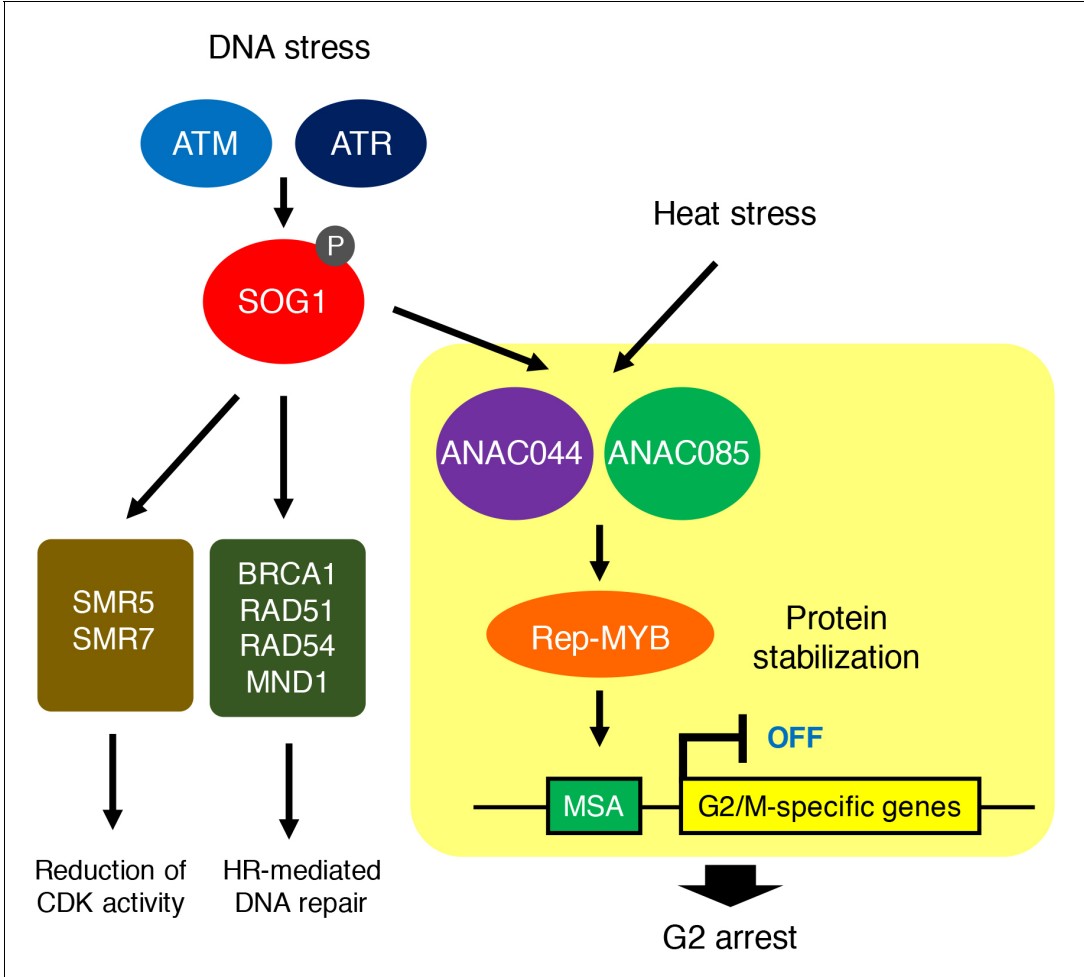

**Figure 10.** ANAC044/ANAC085-mediated pathway involved in stress-induced G2 arrest. Both DNA damage and heat stress induce *ANAC044* and *ANAC085*, which then promote Rep-MYB accumulation and repress a set of G2/M-specific genes, thereby leading to G2 arrest.

DOI: https://doi.org/10.7554/eLife.43944.041

The following figure supplements are available for figure 10:

**Figure supplement 1.** Expression of SOG1 target genes in *anac044 anac085*.

DOI: https://doi.org/10.7554/eLife.43944.042

**Figure supplement 2.** *ANAC044* expression under cold and salt stresses.

DOI: https://doi.org/10.7554/eLife.43944.043

other hand, it is unlikely that ANAC044 and ANAC085 form heterodimers with SOG1 to control the SOG1 target genes, because transcript levels of SOG1 target genes were elevated by bleomycin treatment in *anac044 anac085*, as they were in WT (***Figure 10—figure supplement 1***). However, it remains possible that ANAC044/085 interact with other NAC transcription factors including SOG1, which are induced and/or activated by DNA damage, and control the expression of their target genes.

Here we identified a core signalling module that controls G2 arrest in response to stresses. The promoter region of *ANAC085*, but not *ANAC044*, contains several consensus motifs of the heat shock element, which is recognized by heat shock transcription factors (***Guo et al., 2008***). Moreover, our preliminary results show that *ANAC044* is also up-regulated by cold and salt stresses (***Figure 10—figure supplement 2***). Therefore, we propose that ANAC044 and ANAC085 are induced by distinct stress factors, thereby functioning as a hub in transmitting environmental signals to the cell cycle machinery. ***Kobayashi et al. (2015)*** reported that *Arabidopsis* Rep-MYBs form a protein complex with retinoblastoma-related protein and the E2F transcription factor E2FC, thereby existing

as a component of the DREAM/dREAM-like complex, which has been characterized in vertebrates and *Drosophila* (*Georlette et al., 2007*; *Sadasivam and DeCaprio, 2013*). It is therefore likely that the ANAC044/ANAC085-mediated pathway also controls other subunits of the DREAM complex, thereby differentially controlling downstream genes in response to various stresses. Further studies will reveal whether vertebrate DREAM complexes are also involved in the transition from the mitotic to the quiescent state upon exposure to stresses, and how their activity is regulated to cope with distinct stresses.

# Materials and methods

**Key resources table**

| Reagent type (species) or resource | Designation | Source or reference | Identifiers | Additional information |
|---|---|---|---|---|
| Genetic reagent (*Arabidopsis thaliana*) | *anac044-1* | ABRC stock center | SAIL_1286D02 | homozygous mutant plant line |
| Genetic reagent (*Arabidopsis thaliana*) | *anac044-2* | ABRC stock center | GABI_968B05 | homozygous mutant plant line |
| Genetic reagent (*Arabidopsis thaliana*) | *anac085-1* | ABRC stock center | GABI_894G04 | homozygous mutant plant line |
| Genetic reagent (*Arabidopsis thaliana*) | *anac085-2* | ABRC stock center | SALK_208662 | homozygous mutant plant line |
| Genetic reagent (*Arabidopsis thaliana*) | *anac044-1 anac085-1* | this paper stock center | | homozygous mutant plant line |
| Genetic reagent (*Arabidopsis thaliana*) | *anac044-1 anac085-1 sog1-101* | this paper stock center | | homozygous mutant plant line |
| Genetic reagent (*Arabidopsis thaliana*) | *sog1-101* | *Ogita et al., 2018* | | homozygous mutant plant line |
| Genetic reagent (*Arabidopsis thaliana*) | *ProMYB3R3:MYB3R3-GFP* | *Chen et al., 2017* | | transgenic plant line |
| Genetic reagent (*Arabidopsis thaliana*) | *35S:ANAC044-GR* | this paper | | transgenic plant line |
| Genetic reagent (*Arabidopsis thaliana*) | *Cytrap* | *Yin et al., 2014* | | transgenic plant line |
| Genetic reagent (*Arabidopsis thaliana*) | *myb3r3-1* | *Kobayashi et al., 2015* | | homozygous mutant plant line |
| Genetic reagent (*Arabidopsis thaliana*) | *anac044-1 anac085-1 myb3r3-1* | this paper | | homozygous mutant plant line |
| Genetic reagent (*Arabidopsis thaliana*) | *ProANAC044:GUS* | this paper | | transgenic plant line |
| Genetic reagent (*Arabidopsis thaliana*) | *ProSOG1:SOG1-Myc* | *Yoshiyama et al., 2009* | | transgenic plant line |
| Antibody | rabbit polyclonal αGFP | Thermo Fisher Scientific | A-6455 | 1:3000 for immunoblotting |
| Antibody | rabbit polyclonal αHistone H3 | Abcam | ab1791 | 1:2000 for immunoblotting |
| Antibody | Monoclonal antibody | Millipore | clone 4A6 | 1:500 for chromatin immunoprecipitation |

## Plant materials and growth conditions

*Arabidopsis thaliana* (ecotype Col-0) was used in this study. *sog1-101* (*Ogita et al., 2018*), *ProSOG1:SOG1-Myc* (*Yoshiyama et al., 2013*), *myb3r3-1* (*Chen et al., 2017*), *ProMYB3R3:MYB3R3-GFP* (*Chen et al., 2017*) and the cell cycle marker system Cytrap (*Yin et al., 2014*) were described previously. *anac044-1* (SAIL_1286D02), *anac044-2* (GABI_968B05), *anac085-1* (GABI_894G04) and *anac085-2* (SALK_208662) were obtained from the Arabidopsis Biological Resource Center. To generate *ProANAC044:GUS* and *ProANAC085:GUS*, the 1.5 kb promoters of *ANAC044* and *ANAC085* were PCR-amplified from genomic DNA and cloned into the Gateway entry vector pDONR221 by

BP reaction according to the manufacturer's instructions (Thermo Fisher Scientific). Primer sequences used for PCR are listed in *Supplementary file 1*. An LR reaction was performed with the destination vector pGWB3 (*Nakagawa et al., 2007*) to generate a binary vector carrying the fusion construct with *GUS*. The construct was transferred into the *Agrobacterium tumefaciens* GV3101 strain harbouring the plasmid pMP90 (*Koncz and Schell, 1986*), and the obtained strain was used to generate stably transformed *Arabidopsis* by the floral dip transformation method (*Clough and Bent, 1998*). Plants were grown in Murashige and Skoog (MS) medium under continuous light conditions. For root growth analysis, five-day-old seedlings grown on MS plates were transferred to MS medium containing mannitol or the following DNA-damaging agents: bleomycin (Wako), MMS (Wako), MMC (Nacalai Tesque) or HU (Nacalai Tesque). Seedlings were grown vertically, and the position of root tips was marked every 24 hr. Root growth was measured using ImageJ software by calculating the distance between successive marks along the root axis.

## Quantitative RT-PCR

Total RNA was extracted from *Arabidopsis* roots with a Plant Total RNA Mini Kit (Favorgen). First-strand cDNAs were prepared from total RNA using ReverTra Ace (Toyobo) according to the manufacturer's instructions. Quantitative RT-PCR was performed with a THUNDERBIRD SYBR qPCR Mix (Toyobo) with 100 nM primers and 0.1 µg of first-strand cDNA. PCR reactions were conducted with the Light Cycler 480 Real-Time PCR System (Roche) under the following conditions: 95°C for 5 min; 60 cycles at 95°C for 10 s, 60°C for 20 s and 72°C for 30 s. Primer sequences are listed in *Supplementary file 1*.

## GUS assay

Seedlings were incubated in GUS staining solution [100 mM sodium phosphate, 1 mg/ml 5-bromo-4-chloro-3-indolyl ß-D-glucuronide, 0.5 mM ferricyanide and 0.5 mM ferrocyanide (pH 7.4)] at 37°C in the dark. The samples were cleared with a transparent solution [chloral hydrate, glycerol and water (8 g: 1 ml: 1 ml)] and observed under a light microscope, Axioskop 2 Plus (Zeiss) or SZX16 (Olympus).

## Chromatin immunoprecipitation

ChIP was performed as described previously (*Gendrel et al., 2005*). *pSOG1:SOG1-MYC* seeds were germinated in 100 ml of liquid MS medium, and cultured under continuous light at 23°C with gentle shaking (50 rpm). After a 2 week culture period, bleomycin was added to the medium to a final concentration of 0.6 µg/ml, and the seedlings were cultured for 12 hr. Chromatin bound to the SOG1-Myc fusion protein was precipitated with anti-Myc antibody (Millipore). ChIP-qPCR was performed using immunoprecipitated DNA. Three independent ChIP experiments were conducted. To quantify the precipitated chromatin, gene-specific primers listed in *Supplementary file 1* were used for real-time qPCR. PCR reactions were conducted with the Light Cycler 480 Real-Time PCR System (Roche) under the following conditions: 95°C for 5 min; 60 cycles at 95°C for 10 s, 60°C for 20 s and 72°C for 30 s.

## Microarray

Microarray analysis was carried out as described previously (*Ogita et al., 2018*). Five-day-old seedlings grown on MS plates were transferred onto MS medium containing 0.6 µm/ml bleomycin and grown for 10 hr. Total RNA was extracted from root tips with a Plant Total RNA Mini Kit (Favorgen). Cyanine-3 (Cy3)-labelled cDNA obtained from total RNA was hybridized to an Agilent-034592 Arabidopsis Custom Microarray. Slide scanning was performed using the Agilent DNA Microarray Scanner (G2539A ver. C).

## GUS recombination assay

The *GUS* direct-repeat recombination reporter line #1406 (*Gherbi et al., 2001*) was crossed with *anac044-1 anac085-1* and *sog1-101*. Plants homozygous for the reporter and the alleles of *anac044-1 anac085-1* or *sog1-101* were grown for two weeks, and irradiated with gamma rays (50 Gy). After a three-day cultivation, GUS staining was performed, and the number of blue spots was counted on leaves.

## Pulse labelling with EdU

Five-day-old seedlings were grown on MS plates supplemented with or without 0.6 µg/ml bleomycin for 12 hr. Seedlings were then transferred to liquid MS medium with or without bleomycin and with 20 µM EdU, followed by a 15 min incubation. After washing with MS medium, seedlings were transferred again to MS medium supplemented with or without bleomycin. EdU staining was conducted with a Click-iT Plus EdU Alexa Fluor 488 Imaging Kit (Thermo Fisher Scientific) according to the manufacturer's instructions, and nuclei were stained with DAPI.

## Immunoblotting

Total protein extracted from root tips was separated by 10% sodium dodecyl sulphate-polyacrylamide gel electrophoresis and blotted onto a polyvinylidene difluoride membrane (Millipore). Immunoblotting was conducted with anti-GFP antibody (A-6455, Thermo Fisher Scientific) or anti-histone H3 antibody (ab1791, Abcam) at a dilution of 1:3000 or 1:2000, respectively. Clarity Western ECL Substrate (Bio-Rad Laboratories) was used for detection.

## Acknowledgements

We thank Seiichi Toki for the *GUS* recombination reporter line and Tsuyoshi Nakagawa for the pGWB3 vector. The γ-ray irradiation experiment was supported by the Radiation Biology Center Cooperative Research Program (Kyoto University). This work was supported by MEXT KAKENHI (Grant numbers 17H03965, 17H06470, 17H06477, 17K15141 and 16H01243).

## Additional information

### Funding

| Funder | Grant reference number | Author |
|---|---|---|
| Ministry of Education, Culture, Sports, Science, and Technology | 17H03965 | Masaaki Umeda |
| Ministry of Education, Culture, Sports, Science, and Technology | 17H06470 | Masaaki Umeda |
| Ministry of Education, Culture, Sports, Science, and Technology | 17H06477 | Masaaki Umeda |
| Ministry of Education, Culture, Sports, Science, and Technology | 17K15141 | Naoki Takahashi |
| Ministry of Education, Culture, Sports, Science, and Technology | 16H01243 | Naoki Takahashi |

The funders had no role in study design, data collection and interpretation, or the decision to submit the work for publication.

### Author contributions

Naoki Takahashi, Conceptualization, Data curation, Formal analysis, Funding acquisition, Validation, Investigation, Visualization, Methodology, Writing—original draft, Writing—review and editing; Nobuo Ogita, Tomonobu Takahashi, Shoji Taniguchi, Data curation, Formal analysis, Validation, Investigation, Visualization, Methodology; Maho Tanaka, Data curation, Formal analysis, Validation, Investigation, Methodology; Motoaki Seki, Data curation, Investigation; Masaaki Umeda, Conceptualization, Resources, Data curation, Formal analysis, Supervision, Funding acquisition, Validation, Investigation, Visualization, Methodology, Writing—original draft, Project administration, Writing—review and editing

Author ORCIDs
Naoki Takahashi http://orcid.org/0000-0002-9526-277X
Masaaki Umeda http://orcid.org/0000-0003-3934-7936

Decision letter and Author response
Decision letter https://doi.org/10.7554/eLife.43944.049
Author response https://doi.org/10.7554/eLife.43944.050

## Additional files

### Supplementary files

• Supplementary file 1. Primers used for cloning, qRT-PCR, ChIP-qPCR and semi-quantitative RT-PCR.
DOI: https://doi.org/10.7554/eLife.43944.044

• Transparent reporting form
DOI: https://doi.org/10.7554/eLife.43944.045

### Data availability

Microarray data have been deposited in GEO under accession codes GSE123315.

The following dataset was generated:

| Author(s) | Year | Dataset title | Dataset URL | Database and Identifier |
|---|---|---|---|---|
| Ogita N, Takahashi N, Tanaka M | 2018 | Transcriptomic analysis of Arabidopsis anac044 anac085 and sog1 mutant under DNA damage condition. | https://www.ncbi.nlm.nih.gov/geo/query/acc.cgi?acc=GSE123315 | NCBI Gene Expression Omnibus, GSE123315 |

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
