## [Decision Letter]

Thank you for submitting your article "A regulatory module controlling stress-induced cell cycle arrest in *Arabidopsis*" for consideration by *eLife*. Your article has been reviewed by three peer reviewers, and the evaluation has been overseen by a Reviewing Editor and Christian Hardtke as the Senior Editor.

The reviewers have discussed the reviews with one another and the Reviewing Editor has drafted this decision to help you prepare a revised submission.

The three reviewers greatly valued your discoveries on how specific a stress condition, such as heat stress, triggers cell cycle arrest. They furthermore concluded that the work is suitable for *eLife*. However, we would like you to still experimentally assess the cell numbers after heat stress (essential for revision).

Reviewer #1:

The authors describe a novel regulatory module, which suppresses the progress of G2/M phase in response to DNA damage repair (DDR) and heat shock stress in *Arabidopsis*. In the previous study, the authors found that promoter regions of two ANAC transcription factors, *ANAC044* and *ANAC085*, are bound the SOG1. SOG1 is functional ortholog of the animal p53 protein and an integrator of plant DNA damage repair responses. Phylogenetic analysis revealed that both *ANAC044* and *ANAC085* TFs are closely related to SOG1, suggesting a possible functional relationship. However, both proteins lack the SQ phosphorylation sites typical for the genes activated by phosphorylation signal during DDR. However, both *ANAC044/085* contain SOG1 binding sites in their promoters and are transcriptionally activated in response to DSB inducer bleocin in WT but in *sog1-101* plants. This suggests that *ANAC044/085* are transcriptionally regulated by SOG1. *anac044* and *anac085* mutant plants showed increased tolerance towards several DNA damaging agents and their double mutant repeated the single mutant phenotype. Analysis of the triple mutant with *sog1-101* revealed that both factors are epistatic to SOG1. However, the core DDR genes were transcriptionally up-regulated and HR frequency reached WT-like levels in the *anac044/085* single and double mutants, indicating that they are not involved directly in the repair process. Instead, the *ANAC044/085* seem to influence progression of the cell cycle by enhancing accumulation of the R1R2R3-type Myb transcription factor (Rep-MYB), which represses G2/M-specific genes.

The authors also show that the same module controls cell cycle progression in response to extended heat stress (37 C for 24h) but not osmotic stress.

This is timely study, which characterizes functions of two previously unannotated genes and places them into specific DDR signalling pathway.

The experiments are performed in solid way and have all necessary controls.

The text is clear and easy to read.

I have only one major comment. I am puzzled by the connection between ANAC044/085 and heat stress. I wonder if this is genuine heat stress signalling response or the treatment (far from natural dose for *Arabidopsis*) produces some kind of DNA damage, which then triggers the DDR response. Are the authors able to distinguish between these options by specific experiments?

Here are specific suggestions:

Subsection “*ANAC044* and *ANAC085* are induced by DNA damage” and throughout the manuscript: "generated *promoter::GUS* reporter lines and found that *ProANAC085::GUS* showed”. I suggest using standard nomenclature *Promoter::Gene:Fusion*. It is also recommended using Pro instead of p, which should be reserved for plasmids.

Subsection “SOG1 directly induces *ANAC044* and *ANAC085* under DNA damage conditions”, last paragraph: we conducted ChIP-PCR analysis. Change to qPCR?

The authors could save some space by using more abbreviations: e.g. HR, DDR, *A. thaliana* or *Arabidopsis*.

Subsection “*ANAC044* and *ANAC085* are not involved in homologous recombination-mediated DNA repair”, last paragraph: the authors cite the review on HR system. I would like to suggest using original papers where the principle is described.

"…number of EdU-labelled cells with mitotic figures…" EdU positive cells in interphase were not counted? This is not very clear. Please describe in more detail and/or provide images.

"In contrast, *ANAC044* and *ANAC085* lack the SQ motif, raising the possibility that they do not require phosphorylation for their transcriptional function." Could they contain other than SQ phosphorylation site?

"The GUS recombination reporter line #1406." Please describe which type of the HR construct this reporter line contains.

Figures: It would be great to show also plants (not only graphs) for the Figures 3 and 6B, 7C and 8C.

Reviewer #2:

The SOG1 transcription factor is a key regulator in the plant DNA damage response (DDR) pathway, controlling expression of DNA repair genes and cell cycle checkpoint regulators. In this work, two related proteins (*ANAC044* and *ANACO85*) are demonstrated to work immediately downstream of SOG1. Differently from SOG1 they appear to specifically control (a late) G2 arrest through (indirect) control of the repressive MYB3R3 protein, likely through proteolytic control. *ANAC044* (and *ANACO85*) is as well linked to a heat stress induced G2 arrest, here working independently from SOG1.

The work represents a truly and interesting expansion of the current knowledge on the SOG1 pathway, illustrating a bifurcation of the DDR pathway into a pathway controlling DNA repair and one pathway responsible for a G2 arrest. All experiment are well executed and the manuscript is well written. I only have some doubts on the heat stress data:

1) The difference in growth between control and *anac044 anac085* mutants is only marginal, being about 4 mm over a period of 4 days (Figures 7C and 8C). That is a difference of 1mm/day (control plants grow about 10 mm/day). Can't a different setup be found in which the difference becomes more convincing?

2) I also miss cellular data showing that under these conditions the difference is due to a difference in meristematic cell number and not cell size (thus truly linked with a cell cycle checkpoint control).

Having said this, the manuscript holds more than sufficient interesting data to stand without the heat stress data.

Reviewer #3:

Background: Plants exhibit elaborate and robust regulatory responses to DNA double strand breaks, affecting repair, programmed cell death, and regulation of the cell cycle. SOG1 is a transcriptional activator that regulates virtually all of the rapid transcriptional activation arm of this response. (SOG1 is activated post-translationally through phosphorylation by the damage-detectors ATM and ATR). rep-Myb3R proteins (which repress the expression of G2/M genes) are responsible for most of the transcriptional repression arm, which arises more later and is focused on the repression of the G2/M transition. The regulation of these rep-Myb proteins is only partly understood. It has been established that CDK's routinely promote the degradation of these proteins in G2, and so one proposed mechanism for their regulation in DDR is through the transcriptional induction (by activated SOG1, which directly targets the promoters of these genes) of CDK inhibitors. The authors here state that "…the induction is not sufficient to lead to cell cycle arrest…" and so they further investigate the mechanism of this arrest.

Results: The authors convincingly demonstrate that the SOG1-induced (and closely related) NAC domain transcription factors NAC 044 and 085 are required for damage-induced arrest in G2, and also for SOG1-dependent programmed cell death (as well at SOG1-independent DNA damage-induced mitotic cell death- this is a new function), but not for SOG1-dependent induction of HR. They also show that these TF's are required for the accumulation of MYB3R3 protein in response to DNA damage (though, interestingly, not for the accumulation of MYB3R3 transcript).

They then go on to demonstrate that the transcriptional upregulation of NAC044 and NAC085-and their induction of G2 arrest is recapitulated, without SOG1-dependence, in the response to heat. Thus NAC044 and NAC085 govern a G2-arrest module that can be activated independently by a variety of stresses. The mode of action through which NAC044 and 085 stabilize repMYB3R remains a matter of speculation, but possible pathways are discussed. Given the wide range of assays and phenomenologies discussed here, the paper is remarkably clear.

---

## [Author Response]

[…] I have only one major comment. I am puzzled by the connection between ANAC044/085 and heat stress. I wonder if this is genuine heat stress signalling response or the treatment (far from natural dose for Arabidopsis) produces some kind of DNA damage, which then triggers the DDR response. Are the authors able to distinguish between these options by specific experiments?

We distinguished these two possibilities using the *sog1* mutant. As shown in Figure 8C, *sog1-101* exhibited root growth retardation in response to heat stress to the same extent as wild-type, suggesting that SOG1 is not involved in heat stress-triggered growth arrest. Moreover, *ANAC044* and *ANAC085* were induced normally in *sog1-101* by heat stress, indicating that SOG1 is not required for their induction (Figure 8E, F). Since SOG1 is the central regulator that transmits DNA damage signals, we therefore think that DNA damage is not the cause of cell cycle arrest when plants are exposed to heat stress.

Here are specific suggestions:Subsection “ANAC044 and ANAC085 are induced by DNA damage” and throughout the manuscript: "generated promoter::GUS reporter lines and found that ProANAC085::GUS showed”. I suggest using standard nomenclature Promoter::Gene:Fusion. It is also recommended using Pro instead of p, which should be reserved for plasmids.

We have changed the nomenclature to *Promoter:Gene-Fusion* (e.g., *ProANAC085:GUS, ProSOG1:SOG1-Myc, ProMYB3R3:MYB3R3-GFP*) throughout the manuscript.

Subsection “SOG1 directly induces ANAC044 and ANAC085 under DNA damage conditions”, last paragraph: we conducted ChIP-PCR analysis. Change to qPCR?

We have changed 'ChIP-PCR' to 'ChIP-qPCR'.

The authors could save some space by using more abbreviations: e.g. HR, DDR, *A. thaliana* or *Arabidopsis*.

We have used abbreviations for a few additional words (e.g., HR, DDR, *Arabidopsis*).

Subsection “ANAC044 and ANAC085 are not involved in homologous recombination-mediated DNA repair”, last paragraph: the authors cite the review on HR system. I would like to suggest using original papers where the principle is described.

Thank you for your suggestion. We have cited the original paper describing the HR system (Swoboda et al., 2005).

"…number of EdU-labelled cells with mitotic figures…" EdU positive cells in interphase were not counted? This is not very clear. Please describe in more detail and/or provide images.

We have explained this experiment in more detail in the revised manuscript.

"In contrast, ANAC044 and ANAC085 lack the SQ motif, raising the possibility that they do not require phosphorylation for their transcriptional function." Could they contain other than SQ phosphorylation site?

Our search using NetPhos 3.1 identified several other putative phosphorylation sites in both *ANAC044* and *ANAC085*. However, we cannot readily rely on such programs, and thus have not added any statement in the revised manuscript. Please note that we mentioned the possibility of phosphorylation by other kinase(s) in the original text.

"The GUS recombination reporter line #1406." Please describe which type of the HR construct this reporter line contains.

We used the *GUS* direct-repeat recombination reporter line (Swoboda et al., 2005). We have indicated this information.

Figures: It would be great to show also plants (not only graphs) for the Figures 3 and 6B, 7C and 8C.

We have shown the plants in supplementary figures: Figure 2—figure supplement 3, Figure 7—figure supplement 1, Figure 8—figure supplement 2, and Figure 9—figure supplement 2.

Reviewer #2:[…] 1) The difference in growth between control and anac044 anac085 mutants is only marginal, being about 4 mm over a period of 4 days (Figures 7C and 8C). That is a difference of 1mm/day (control plants grow about 10 mm/day). Can't a different setup be found in which the difference becomes more convincing?

We tested several heat stress conditions, such as 37 ^o^C incubation for 3 h, 6 h, 12 h, 24 h and 24 h. However, 24-h treatment showed the most significant difference between WT and the mutants. We might be able to observe a bigger difference by continuous incubation at 37 ^o^C, but such a stressful condition would probably cause side effects on root growth. Therefore, we have left the original data in the revised manuscript.

2) I also miss cellular data showing that under these conditions the difference is due to a difference in meristematic cell number and not cell size (thus truly linked with a cell cycle checkpoint control).

Thank you for your suggestion. We counted the meristematic cell number in wild-type, *anac044-1 anac085-1, myb3r3-1* and *anac044-1 anac085-1 myb3r3-1* after heat stress. The result showed a lower sensitivity of the tested mutants to heat stress in terms of reduction of the meristematic cell number. We have included these data as Figure 8D and explained the result in the revised manuscript.